# Src-dependent tyrosine-phosphorylation of NM2A has a protective role against bacterial pore-forming toxins

Cláudia Brito[1,2,3¤a‡], Francisco S. Mesquita[1¤b‡], Joana M. Pereira[1,2,3¤c‡],
Daniel S. Osório[1], Neil Billington[4], Ricardo R. Lima[1], Sílvia Vale-Costa[1,3],
James R. Sellers[4], Didier Cabanes[1], Ana X. Carvalho[1,5]*, Sandra Sousa◉[1,3]*

1 i3S-Instituto de Investigação e Inovação em Saúde, Universidade do Porto, Porto, Portugal,
2 MCBiology PhD Program– ICBAS, Universidade do Porto, Porto, Portugal, 3 IBMC, Instituto de Biologia Molecular e Celular, Universidade do Porto, Porto, Portugal, 4 Cell and Developmental Biology Center, National Heart Lung and Blood Institute, NIH, Bethesda, Maryland, United States of America, 5 Biology Department, Faculty of Sciences, University of Porto, Porto, Portugal

‡ Shared co-first authorship.
¤a Quantitative Cell Biology Department, Centre for Genomic Regulation (CRG), Barcelona, Spain,
¤b Lee Kong Chian School of Medicine, Nanyang Technological University, Singapore, Singapore,
¤c Chromatin and Infection Laboratory, Institut Pasteur, Paris, France
* anacarvalho@i3s.up.pt (AXC); srsousa@i3s.up.pt (SS)

## Abstract

Pore-forming toxins (PFTs) are key bacterial virulence factors that disrupt host plasma membrane (PM) integrity, triggering cellular stress and initiating repair mechanisms. The cytolysin Listeriolysin O (LLO), secreted by *Listeria monocytogenes*, has well established roles in infection, yet the host signaling responses to LLO-induced damage remain poorly understood. Here, we identify a previously unrecognized protective pathway in which LLO triggers rapid activation of the tyrosine kinase Src, leading to phosphorylation of the non-muscle myosin II heavy chain 2A (NMHC2A) at tyrosine 158. While Src activation and NMHC2A tyrosine phosphorylation have been observed during *Listeria* infection, we demonstrate here that both responses are directly driven by LLO. This phosphorylation event does not alter NMHC2A motor activity *in vitro* but is required for cytoskeletal reorganization and efficient responses to PM damage. Using *Caenorhabditis elegans*, we further show that phosphorylation of the NMHC2A homolog NMY-2 at the conserved tyrosine 163 is required for survival under PFT-induced stress and heat shock, revealing an evolutionarily conserved defense mechanism. Together, our findings establish Src-mediated NMHC2A phosphorylation as a critical link between PFT-induced PM damage sensing and actomyosin remodeling, advancing our understanding of host defense against bacterial toxins.

**Data availability statement:** Full Western blot images and corresponding quantifications will be made publicly available via Zenodo (https://zenodo.org/records/18255578). Additionally, all raw data used to generate the graphs in the manuscript will also be deposited and accessible through Zenodo (https://zenodo.org/records/18255578). All other relevant data are in the manuscript and its Supporting information files.

**Funding:** This work was funded by Operation no. 15828 – COMPETE2030-FEDER-00693900 supported by PITD, Portugal 2030, and the European Union, by FEDER—Fundo Europeu de Desenvolvimento Regional funds through the COMPETE 2020—Operational Programme for Competitiveness and Internationalization (POCI), Portugal 2020, by Portuguese funds through FCT - Fundação para a Ciência e a Tecnologia/Ministério da Ciência, Tecnologia e Ensino Superior in the framework of the project POCI-01-0145-FEDER-030863 (PTDC/BIA-CEL/30863/2017), and by the European Research Council under the European Union's Horizon 2020 Research and Innovation Programme (grant agreement 640553 – ACTOMYO - to AXC). CB, FSM and JMP were supported by FCT fellowships (SFRH/BD/112217/2015, SFRH/BPD/94458/2013 and SFRH/BD/143940/2019 respectively). CB was a Fulbright and FLAD fellow. JMP received funds from Fulbright, EMBO Scientific Exchange Grant (number 10182), and FLAD. SVC, SS and AXC received support from the FCT Institutional and Individual CEEC program, respectively (CEECINST/00091/2018, 2022.04457.CEECIND and CEECIND/01967/2017). AXC also received support from FCT-TENURE first edition (2023.14140.TENURE.013). The authors acknowledge the support of Advanced Light Microscopy i3S Scientific Platform, member of the national infrastructure PPBI-Portuguese Platform of BioImaging (supported by POCI-01-0145-FEDER-022122). The funders had no role in study design, data collection and analysis, decision to publish, or preparation of the manuscript.

**Competing interests:** The authors have declared that no competing interests exist.

## Author summary

Pathogenic bacteria often produce toxins that create pores in the membranes of host cells, compromising cell survival and helping the infection spread. To counter this, host cells must quickly detect and repair the damage. We studied how cells respond to Listeriolysin O (LLO), produced by the human pathogenic bacterium *Listeria monocytogenes*. We found that LLO activates the host Src kinase, which in turn modifies a key structural protein, non-muscle myosin II (NMII). Although this modification does not affect NMII's normal motor function, it is crucial for organizing the cellular skeleton and protecting the cell's membrane after toxin damage. Importantly, we found that this protective mechanism is conserved in the worm *Caenorhabditis elegans*, where it also defends against a similar bacterial toxin and even heat stress. Our findings reveal a new way that cells use existing structural proteins to rapidly respond to external threats, providing insight into how organisms protect themselves from harmful microbes and environmental stress.

## Introduction

Pore-forming toxins (PFTs) are widespread bacterial virulence factors that compromise host cell integrity by creating transmembrane pores [1]. This disrupts ion homeostasis, triggers abnormal intracellular signaling, and ultimately leads to cell death [2]. PFTs contribute to bacterial pathogenesis by promoting tissue inflammation, breaching epithelial barriers, and impairing immune defenses, thus facilitating bacterial spread and disease progression [3]. Host cells must rapidly detect plasma membrane (PM) disruption and activate repair mechanisms to restore cellular integrity and homeostasis [2]. These responses often involve dynamic remodeling of the cortical cytoskeleton, which generates contractile forces essential for PM resealing, bleb formation and retraction, and the release of extracellular vesicles [2,4,5].

*Listeria monocytogenes* secretes Listeriolysin O (LLO), a cholesterol-dependent cytolysin with multiple roles in infection [6]. LLO facilitates bacterial escape from the phagosome, enabling its intracellular survival, and contributes to immune evasion [7]. Emerging evidence suggests that, beyond causing PM disruption, LLO can activate host signaling pathways [8,9], including those involving tyrosine kinases and cytoskeletal effectors, that are implicated in PM repair and cellular recovery [4,10]. However, the molecular mechanisms coordinating these protective responses remain to be fully elucidated.

The non-muscle myosin 2A (NM2A) is a key effector of cytoskeletal organization and mechanical force generation [11,12]. NM2A is a hexameric complex comprising two non-muscle myosin heavy chains (NMHC2A) and two pairs of light chains: the regulatory (RLCs) and the essential (ELCs) light chains [11]. The NMHC2A subunits contain both the ATPase activity and the actin-binding site, enabling the generation of cortical contractile forces that regulate focal adhesions, cell migration, and cell shape

[11,13]. NMHC2A has also been implicated in responses to mechanical and osmotic stress [14,15], as well as in the repair of PFT-induced PM damage [4,16]. Previous studies have shown that NMHC2A undergoes tyrosine phosphorylation in response to diverse bacterial infections [17]. Notably, *L. monocytogenes* infection induces Src-dependent phosphorylation of NMHC2A at Tyr[158] [17].

Src is a non-receptor tyrosine kinase that transduces signals from the extracellular environment to the cytoskeleton, thereby regulating cell adhesion, migration, and PM repair [18]. Src is activated during *L. monocytogenes* infection and modulates both bacterial entry and host cell responses [17,19]. However, it remains unclear whether LLO alone is sufficient to activate Src and whether Src activity directly regulates NMHC2A phosphorylation and cytoskeletal remodeling in response to pore formation.

To better understand the cellular mechanisms responding to PFTs during infections, we investigated here the molecular pathway linking LLO intoxication to Src kinase activation and NMHC2A phosphorylation. We show that LLO is sufficient to trigger rapid and sustained Src activation at the PM, leading to the phosphorylation of NMHC2A at Tyr[158]. Although this modification does not alter NM2A motor activity *in vitro*, it is essential for coordinating cytoskeletal reorganization and protecting PM integrity *in vivo*. Using *Caenorhabditis elegans* as a model, we further demonstrate that phosphorylation of the NMY-2, a homolog of the human non-muscle myosin II heavy chains (NMHC2), at the conserved Tyr[163] residue is required for worm survival in response to PFT exposure and heat stress. Our findings define an evolutionarily conserved mechanism by which NMHC2A phosphorylation, likely mediated by Src kinase, promotes host defense against PM damage by triggering cytoskeletal remodeling.

## Results

### LLO intoxication induces tyrosine phosphorylation of NMHC2A and activation of Src kinase

Given that *Listeria* infection activates Src kinase [19] and leads to Src-dependent phosphorylation of NMHC2A Tyr[158] [17], and because its pore-forming toxin LLO remodels the NM2A cytoskeleton [4,10], we asked whether LLO is necessary and sufficient to trigger NMHC2A tyrosine phosphorylation and activate Src kinase. Using an anti-pTyr antibody, the total tyrosine phosphorylated proteins were immunoprecipitated (IP) from lysates of HeLa cells under control non-intoxicated conditions (NI) or after incubation with a sub-lytic dose of purified LLO (0.5 nM, 15 min), a condition that allows PM damage repair and prevents cell death [5]. While the levels of NMHC2A in whole cell lysates (WCL) were comparable in NI and LLO-intoxicated samples, NMHC2A was significantly enriched in pTyr IP fractions from LLO-intoxicated cells (Fig 1A and 1B). Similar results were obtained in HeLa cells infected with the wild-type (WT) *L. monocytogenes* strain (1h, MOI 200) but not in cells infected with the isogenic strain lacking LLO (ΔLLO) (S1A and S1B Fig). These indicate that LLO is necessary and sufficient to trigger NMHC2A tyrosine phosphorylation. To further extend our observations, we assessed if other LLO-related bacterial PFTs [1], such as perfringolysin O (PFO) produced by *Clostridium perfringens* [16] and pneumolysin (PLY) released by *Streptococcus pneumoniae* [3], also trigger NMHC2A tyrosine phosphorylation. We found that both PFO and PLY trigger the tyrosine phosphorylation of NMHC2A upon intoxication of HeLa cells (S2A-S2D Fig). However, we showed that NMHC2A phosphorylation is impaired upon intoxication of HeLa cells with a pore-deficient PFO variant (PFO inactive) or with LLO in calcium-free medium, demonstrating that this event depends on pore formation and the consequent influx of calcium (S2A and S2B Fig). These findings indicate that NMHC2A tyrosine phosphorylation represents a conserved host cell response to toxin-induced pore formation.

To determine whether LLO-induced NMHC2A phosphorylation correlates with Src activation, we evaluated the levels of activated Src in non-intoxicated (NI) and in LLO-intoxicated HeLa cells. Full activation of Src requires both dephosphorylation of its tyrosine 530 (Tyr[530]) and phosphorylation of its tyrosine 419 (Tyr[419]) [20,21], which can be detected by immunoblot. We found that the levels of Src phosphorylated on Tyr[419] (Src pTyr[419]) and dephosphorylated on Tyr[530] (active Src) were increased upon 5 min-LLO intoxication of HeLa cells (Figs 1C, 1D, S1C and S1D). Activation of Src kinase was also detected in HeLa cells intoxicated by PLY, suggesting that Src signaling is a common response to various PFTs

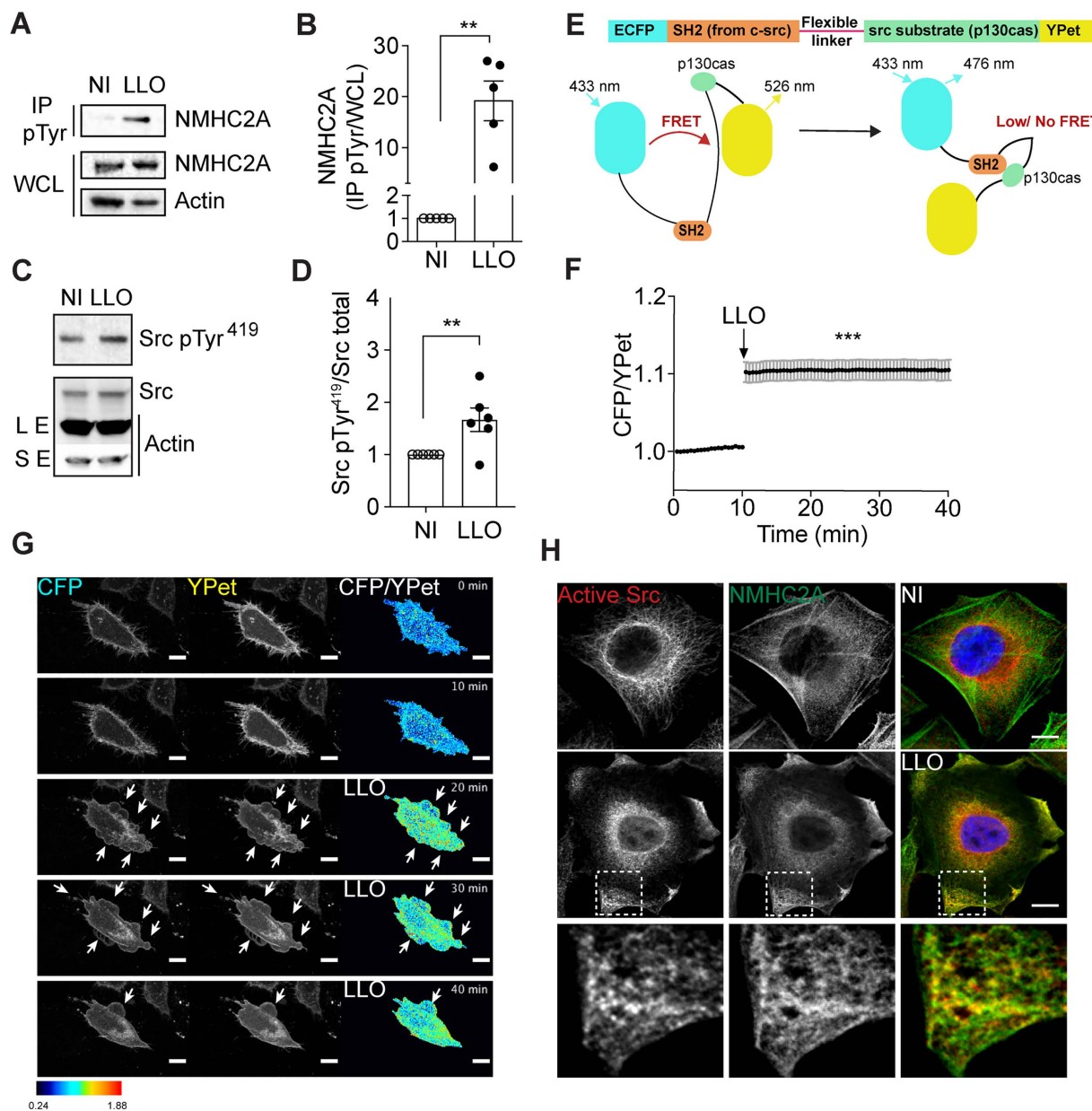

**Fig 1. LLO promotes NMHC2A tyrosine phosphorylation and Src activation in HeLa cells.** (A, B) Levels of NMHC2A measured by immunoblots on whole-cell lysates (WCL) and immunoprecipitated (IP) fractions of pTyr proteins (IP pTyr) from non-intoxicated (NI) or LLO-intoxicated (LLO; 0.5 nM, 15 min) HeLa cells. Actin was used as loading control. (B) Levels of NMHC2A in the IP pTyr fraction (NMHC2A pTyr) were quantified and normalized to those detected in the WCL (NMHC2A WCL). Each dot corresponds to an independent experiment. Data correspond to mean ± SEM ($n = 5$); $p$-value was calculated using two-tailed unpaired Student's $t$-test, **$p < 0.01$. (C) Levels of Tyr419-phosphorylated Src (Src pTyr419) and total Src (Src Total) assessed by immunoblots on WCL of HeLa cells non-intoxicated (NI) or LLO-intoxicated (0.5 nM LLO, 5 min). Actin was used as loading control, long (LE) and short (SE) exposures are shown. (D) Levels of Src pTyr419 signals were quantified and normalized to the levels of total Src. Each dot corresponds to an independent experiment. Data correspond to mean ± SEM ($n = 6$); $p$-values were calculated using two-tailed unpaired Student's $t$-test, **$p < 0.01$. (E) Schematic representation of Src biosensor composed of ECFP, a SH2 domain, a flexible linker, a substrate peptide derived from p130cas, and YPet (22). In conditions where Src is not activated ECFP and YPet are close and FRET occurs. If Src is activated, the SH2 and p130cas domains interact thus increasing the distance between ECFP and Ypet. Consequently, no FRET occurs and the ECFP/YPet ratio increases. (F) Kinetics of donor/acceptor fluorescence emission (ECFP/YPet) of the Src biosensor in NI and LLO-intoxicated cells. LLO was added to the culture medium 10 min after the acquisition started. Values correspond to the mean ± SEM ($n > 30$); $p$-values were calculated using one-way ANOVA with Dunnett's *post hoc* analysis, ***$p < 0.001$. (G) Sequential frames of time-lapse confocal microscopy FRET analysis of NI or LLO-intoxicated HeLa cells expressing ECFP/YPet-based Src biosensor. LLO was added to the culture medium 10 min after the acquisition started. Arrows indicate LLO-induced plasma membrane (PM) blebbing

sites. Scale bar, 10 µm. (H) Confocal microscopy images of NI or LLO-intoxicated (0.5 nM, 15 min) HeLa cells, immunolabeled for active Src (red) and NMHC2A (green) and stained with DAPI (blue). Insets show LLO-induced cortical accumulations of NMHC2A enriched in active Src. Scale bar, 10 µm.

(S2E and S2F Fig). In addition, activated Src kinase was detected in HeLa cells infected with the wild-type (WT) *L. monocytogenes* strain (10 min, MOI 200) but not in those infected with the ΔLLO strain (S1E and S1F Fig). Altogether, these results indicate that LLO is necessary and sufficient to induce Src activation, which presumably might occur upstream of the tyrosine phosphorylation of NMHC2A.

To further confirm this, we examined the activation of Src in single cells using the specific FRET-based ECFP/YPet KRas-Src biosensor [22]. Briefly, this biosensor contains the c-Src SH2 domain, a flexible linker, and a substrate peptide derived from p130cas, which are concatenated between FRET donor ECFP and acceptor YPet (Fig 1E). This Src biosensor has a high FRET signal at rest, due to a strong YPet emission signal when ECFP is excited. As the activated Src phosphorylates the substrate, the biosensor undergoes conformational changes, resulting in the decrease of FRET and YPet emission signals. Therefore, Src activation status can be visualized and quantified by calculating the ratio of CFP/YPet [22]. HeLa cells expressing the biosensor were imaged starting 10 min before LLO was added and the ratio of CFP/YPet signals was calculated over time. The activation levels of Src remained stable before the addition of LLO (Fig 1F and 1G and S1 Movie). However, quickly upon LLO intoxication, the CFP/YPet ratio increased and remained high over time (Fig 1F and 1G and S1 Movie), reflecting the global and lasting Src activation in LLO-intoxicated cells. Of note, increased CFP/YPet values were also detected associated with the PM, where the Src kinase activity was often reported [23] and where the cell is responding to LLO-induced damage by forming blebs (Fig 1G, arrows) [4]. Concurrently, confocal microscopy analysis of NI and LLO-intoxicated HeLa cells showed that, whereas active Src is predominantly cytoplasmic in NI cells, LLO exposure induces its strong enrichment at the cell cortex, colocalizing with the NMHC2A cortical accumulations that serve as hallmarks of PFT-induced PM damage [4,16] (Fig 1H). Similar observations were made in PLY-intoxicated cells (S2G Fig). These results suggest that Src kinase activity at the cell periphery may regulate cytoskeletal remodeling in response to LLO-induced damage. In addition, these data indicate that LLO is sufficient to activate Src, which in turn may control the tyrosine phosphorylation of NMHC2A, as it occurs during *Listeria* infection [17].

## Src kinase activity drives LLO-induced NMHC2A tyrosine phosphorylation

To determine if Src kinase is necessary for the tyrosine phosphorylation of NMHC2A mediated by LLO, we manipulated Src activity by: 1) pharmacological inhibition using Dasatinib (Dasa), 2) downregulation of Src expression using specific shRNAs (shSrc) [17,24], and 3) overexpression of Src kinase mutants. The use of multiple approaches to modulate Src activity ensures data robustness and mitigates concerns related to inhibitor specificity or potential off-target effects. Levels of NMHC2A were assessed in IP pTyr fractions obtained from NI and LLO-intoxicated cells in control or Src perturbed conditions. In control conditions, LLO consistently induced an enrichment of NMHC2A in the IP pTyr fraction (Fig 2A-2F). In contrast, NMHC2A pTyr was reduced upon Dasatinib treatment (Dasa; Fig 2A and 2B), downregulation of Src kinase expression (shSrc; Fig 2C and 2D) or ectopic overexpression of a Src kinase dead mutant (KD; Fig 2E and 2F). Inversely, upon LLO intoxication, the ectopic overexpression of a constitutively active (CA) Src mutant increased NMHC2A pTyr to levels higher than those detected in cells expressing wild-type Src (WT) (Fig 2E and 2F). Notably, low expression levels of Src CA and Src KD when compared to that of WT Src were sufficient to modulate the levels of NMHC2A pTyr (Fig 2E and 2F). These results show that the tyrosine phosphorylation of NMHC2A triggered by LLO depends on the activation of Src kinase.

We previously demonstrated that Src-dependent phosphorylation of NMHC2A triggered by *Listeria* infection, occurs at tyrosine residue in position 158 (Tyr[158]) [17]. Given our findings described above, we hypothesized that LLO is the bacterial factor that drives the Src-dependent phosphorylation of NMHC2A Tyr[158]. To verify this, HeLa cells, either NI or

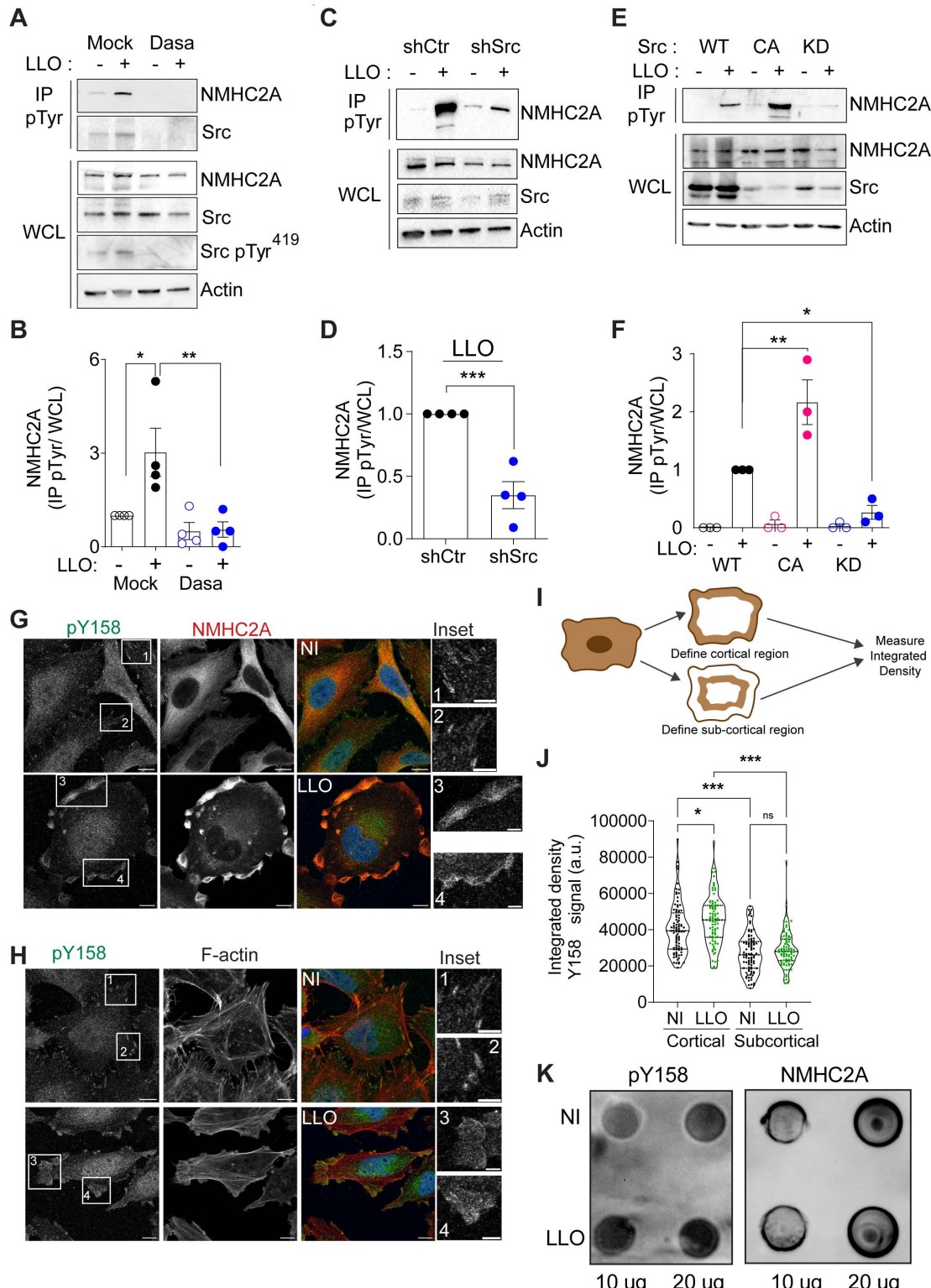

**Fig 2. LLO-induced tyrosine phosphorylation of NMHC2A requires Src activity and occurs in Tyr[158]. (A, C, E)** Immunoblots for NMHC2A, total Src and Src pTyr[419] on WCL or IP pTyr fractions of non-intoxicated (-) or LLO-intoxicated (0.5 nM, 15 min) (+) HeLa cells **(A)** in the absence (Mock) or presence of Dasatinib (300 nM, 1 h) (Dasa); **(C)** expressing control (shCtr) or Src-specific oligos (shSrc), and **(E)** ectopically expressing wild-type Src (WT), constitutively active Src (CA) or kinase dead Src (KD) variants. Actin was used as loading control. **(B, D, F)** Shows the quantified levels of

NMHC2A in IP fractions (NMHC2A pTyr) normalized to those detected in the WCL (NMHC2A WCL), under the different conditions described above. Each dot corresponds to an independent experiment. Values correspond to the mean ± SEM (n ≥ 3) and p-values were calculated using **(B, F)** one-way ANOVA with Dunnett's *post hoc* analysis, *$p < 0.05$, **$p < 0.01$ or **(D)** two-tailed unpaired Student's *t-test*, ***$p < 0.001$. (G, H) Confocal microscopy images of NI or LLO-intoxicated (0.5 nM, 15 min) HeLa cells, immunolabeled for NMHC2A phosphorylated in Tyr[158] (pY158, green) and total NMHC2A (red in G), and stained for F-actin (red in H). DAPI was used to stain the nuclei (blue). Insets in NI cells show defined puncta structures at the cell periphery, which are associated to the extremities of stress fibers detected by F-actin staining (H). The insets in LLO-intoxicated cells show cortical accumulations of NMHC2A phosphorylated in Tyr[158] (pY158, green in **G**, H), total NMHC2A (red in **G**) and F-actin (red in H). Scale bar, 10 μm. (I, J) Levels of NMHC2A phosphorylated in Tyr[158] detected at the cortex of NI and LLO-intoxicated cells. (I) Schematic representation of the protocol used to quantify the pY158 signal specifically at the cortical and sub-cortical regions, detailed in the Material and Methods section. (J) Quantification of Y158 signal (integrated density values) in confocal images similar to those shown in G and H. Values correspond to the mean ± SEM (*n* = 83) and *p*-values were calculated using two-way ANOVA with Tukey's *post hoc* analysis, *$p < 0.05$, ***$p < 0.001$ (K) Dot blots for pY158 NMHC2A and total NMHC2A performed using 10 and 20 μg of total protein extracts from NI or LLO-intoxicated (0.5 nM, 10 min) HeLa cells.

incubated with LLO were fixed and immunostained using an antibody raised against phosphorylated NMHC2A Tyr[158] (anti-pY158) [17]. In NI conditions, the anti-pY158 antibody revealed faint cytoplasmic labeling with defined puncta located at the cell periphery (Fig 2G). We detected pY158 labeling associated with the extremities of stress fibers showed by F-actin staining (Fig 2H). These observations are in agreement with our previous data showing that cells overexpressing a non-phosphorylatable form of NMHC2A (NMHC2A Y158F) display altered focal adhesions [24]. In response to LLO intoxication, the pY158 labeling significantly accumulates at the cell cortex at sites where NMHC2A is also accumulating (Fig 2G) [4] and where actomyosin undergoes remodeling (Fig 2H) [4]. Quantification of the pY158 signal at the cortex of the cell revealed a significant increase in LLO intoxicated cells as compared to NI resting cells (Fig 2I and 2J). The increased phosphorylation in NMHC2A residue Tyr[158] was also confirmed by dot blot using the anti-pY158 (Fig 2K). While the levels of total NMHC2A remain similar in NI and LLO-intoxicated cells (Fig 2K, right panel), the levels of NMHC2A pY[158] are higher in intoxicated cells (Fig 2K, left panel). These data indicate that LLO induced the Src-dependent phosphorylation of NMHC2A in the residue Tyr[158].

## Src kinase coordinates LLO-induced NMHC2A remodeling

Next, we investigated whether Src kinase interferes with the accumulation of cortical NMHC2A induced by LLO, which constitutes a hallmark for PM repair and for the establishment of host survival responses to PFTs [4]. HeLa cells pre-incubated with the Src pharmacological inhibitor Dasatinib (Dasa) [25] or expressing low levels of Src (through specific shRNAs) were challenged with LLO, immunostained for NMHC2A and analyzed by confocal microscopy (Figs 3A and S3A). The percentage of cells displaying LLO-induced cortical accumulations of NMHC2A was not altered by perturbing either Src activity or expression (Fig 3B). However, while the majority of control cells displayed 1 or 2 NMHC2A accumulations, the percentage of cells displaying at least 3 accumulations was significantly increased in Src-deficient cells (Fig 3C and 3D). These data suggest that, while dispensable for the cortical accumulation of NMHC2A induced by LLO, Src activity is necessary to limit the remodeling of the cytoskeleton in response to intoxication. We thus hypothesized that Src activity is critical to restrain PM damage. We measured, by flow cytometry, the PM permeabilization induced by sub-lytic doses of LLO following propidium iodide (PI) incorporation in NI or LLO-intoxicated HeLa cells, under control or Src-deficient conditions. In such conditions LLO intoxication does not cause cell death but repair machinery is activated [5]. As expected, the percentage of PI-positive cells significantly increased after LLO-intoxication (Figs 3E and S4). Despite the moderate differences, this increase was significantly more pronounced in cells with impaired Src activity (Dasa treated) or with low Src expression (shSrc) (Fig 3E), indicating that Src activity limits LLO-induced PM damage. Additionally, we measured, by high-content microscopy, the PM permeabilization induced by increasing concentrations of LLO over time following Draq7 incorporation in NI or LLO-intoxicated HeLa cells, under control or Src-impaired conditions. Draq7 is a non-toxic, far-red fluorescent, membrane-impermeable DNA dye that selectively labels cells with compromised membrane

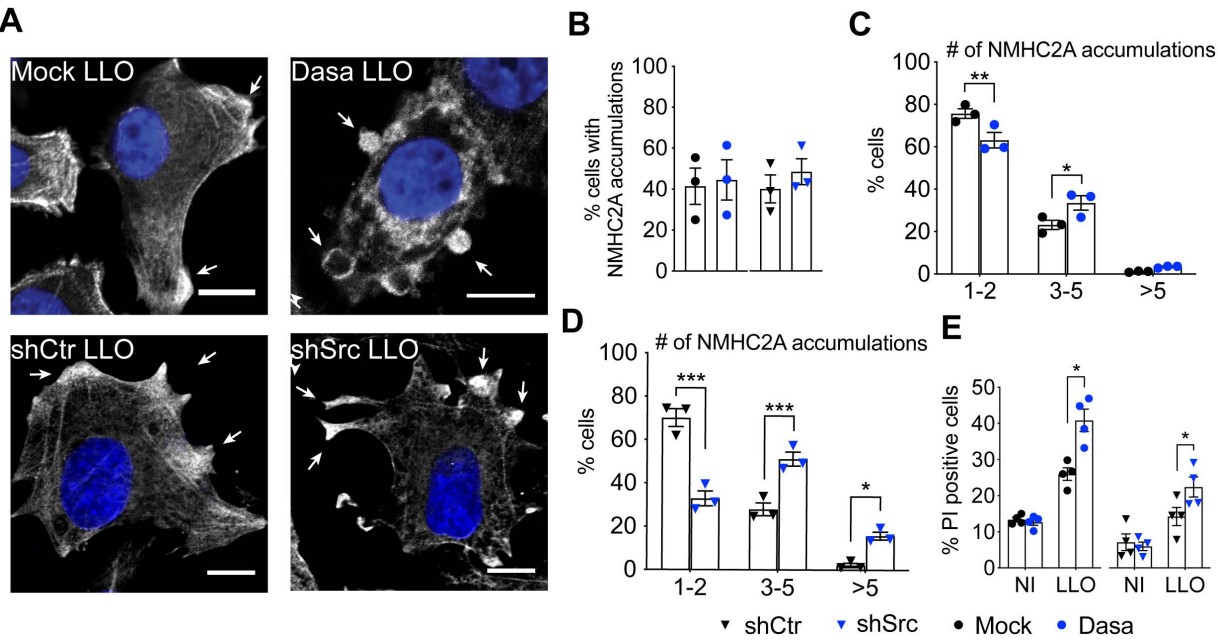

**Fig 3. Src kinase orchestrates the response to LLO-intoxication by controlling actomyosin remodeling in HeLa cells.** (A) Confocal microscopy images of LLO-intoxicated (0.5 nM, 15 min) HeLa cells in control (Mock and shCtr, respectively black circles and black triangles) and Src-impaired (Dasatinib-treated and shSrc, respectively blue circles and blue triangles) conditions. Cells were immunolabeled for NMHC2A (greyscale) and stained with DAPI (blue). White arrows indicate NMHC2A cortical accumulations. Scale bar, 10 µm. (B) Percentage of LLO-intoxicated cells displaying cortical accumulations of NMHC2A under the conditions described in (A). Values are the mean ± SEM (n = 3). (C, D) Percentage of LLO-intoxicated cells showing 1-2, 3-5 or >5 NMHC2A accumulations, under the conditions described in (A). Values are the mean ± SEM (n = 3); p-values were calculated using two-way ANOVA with Sidak's *post hoc* analysis, *p < 0.05, **p < 0.01, ***p < 0.001. (C) Shows data from cells under Src chemical inhibition (Dasa). (D) Shows data from cells in which Src expression was downregulated through shRNAs. (E) Percentage of PI-positive HeLa cells in non-intoxicated (NI) conditions or LLO-intoxicated (LLO) as in (A), measured by flow cytometry analysis. Values are the mean ± SEM (n = 4); p-values were calculated using one-way ANOVA with Dunnett's *post hoc* analysis, *p < 0.05.

integrity, analogous to PI [26]. As before, the percentage of Draq7-positive cells significantly increased over time after intoxication with 0.5 nM and 1 nM of LLO (S3B Fig). This increase was significantly more pronounced in cells with impaired Src activity (Dasa treated). Importantly, the LLO concentrations used did not induce extensive cell death over the duration of the experiment, as the total number of cells in the end of the experiment remained similar across conditions (S3C Fig) and remained unaltered throughout time (S3D Fig), indicating that Draq7 positivity reflects transient PM permeabilization rather than loss of cell viability. These results are consistent with a role for Src activity in regulating the actomyosin cytoskeleton remodeling, and contributing to limiting PM damage upon LLO intoxication, either by restricting pore formation and/or by promoting cellular responses that facilitate pore removal or membrane repair.

### NMHC2A pTyr does not affect the kinetic and structural properties of NM2A *in vitro*

We previously reported that the phosphorylation status of NMHC2A Tyr158 regulates cytoskeletal organization, affects the assembly/disassembly of focal adhesions, and interferes with cell migration [24]. Tyr158 is localized in the proximity of the NM2A ATPase domain [17] (S5A Fig) and is highly conserved across evolution (S5B Fig). These observations led us to hypothesize that the phosphorylation status of Tyr158 may directly interfere with NMHC2A activity. We investigated if phosphorylation of Tyr158 perturbs NM2A ATPase activity or actin-binding capacity, by determining the mechanical and kinetic parameters of human NMHC2A heavy meromyosin (HMM) fragments, which do not form filaments but retain ATPase and actin-binding/translocating activities. We used HMM fragments derived from wild-type NMHC2A (NM2A-GFP-Flag^HMM-WT),

a non-phosphorylatable mutant in which Tyr[158] was substituted by a phenylalanine (Phe, NM2A-GFP-Flag[HMM-Y158F]), and a mutant mimicking a permanent phosphorylation in which Tyr[158] was replaced by a glutamate (Glu, NM2A-GFP-Flag[HMM-Y158E]). Such HMM fragments were co-expressed and co-purified with human RLC and mouse ELC from Sf9 insect cells (S6A Fig). Purified molecules were analyzed by electron microscopy and no strong conformational changes were detected among the different HMM variants (NM2A-GFP-Flag[HMM-WT/Y158F/Y158E], S6B Fig), suggesting that the substitution of Tyr[158] by a Phe or a Glu does not affect the overall HMM structure. The enzymatic activity of the NM2A-GFP-Flag[HMM] variants was measured in a steady-state ATPase assay. Although the mutant molecules appeared to hydrolyze ATP at a slightly higher rate than the WT NM2A-GFP-Flag[HMM] (Fig 4A), this difference was not significant and the $K_{ATPase}$ (concentration of actin required for one-half $V_{max}$) and $V_{max}$ values obtained were similar (Fig 4B), and comparable to those previously described for WT NM2A [27]. This indicates that the tyrosine phosphorylation status of the NMHC2A does not affect the ATPase activity of NM2A. The binding affinity of NM2A-GFP-Flag[HMM] variants to actin was then assessed by co-sedimentation assays in the absence or presence of ATP. Actin alone was pelleted by ultracentrifugation, whereas NM2A-GFP-Flag[HMM] variants remained in the supernatant in the absence of actin (Fig 4C). NM2A-GFP-Flag[HMM] molecules co-sedimented with actin in the absence of ATP but remained in supernatants in the presence of 1 mM ATP (Fig 4C). The fraction of each NM2A-GFP-Flag[HMM] variant bound to actin in the presence of ATP was comparable in all conditions (Fig 4D). Corroborating these data, electron microscopy analysis of mixtures of actin and the different NM2A-GFP-Flag[HMM] variants in the presence or absence of ATP showed similar behaviors (S6C Fig). Finally, we assessed if NMHC2A Tyr[158] is involved in the velocity with which actin filaments slide on NM2A-GFP-Flag[HMM]. We performed *in vitro* actin-gliding assays using coverslip-immobilized NM2A-GFP-Flag[HMM]. All the NM2A-GFP-Flag[HMM] variants translocated actin at similar rates (Fig 4E and S2 Movie), comparable to those previously reported for human NM2A [28].

These results demonstrate that the substitution of the Tyr[158] by a Phe or Glu does not significantly affect the *in vitro* biochemical properties of NM2A, including ATPase activity and actin binding, indicating that NMHC2A pTyr[158] does not directly regulate these intrinsic motor functions under purified conditions. While these findings establish that Tyr158 phosphorylation is dispensable for NM2A motor activity *in vitro*, they do not allow direct assessment of the relevance of these biochemical properties in cells. Thus, our data are consistent with the idea that LLO-induced remodeling of NMHC2A is not driven by changes in ATPase activity but may instead involve regulation of filament assembly and disassembly [24]; this interpretation remains to be fully established.

### Tyrosine phosphorylation of the *C. elegans* NMY-2 is required for survival responses triggered by pore-forming toxins and heat shock

Our data led us to hypothesize that, although phosphorylation of NMHC2A at Tyr[158] does not affect the biochemical properties of NM2A, it may play an important role in cellular responses to infections by PFT-producing bacteria. We therefore used the *C. elegans* model to test whether NMHC2A pTyr[158] contributes to organismal resistance to PFTs.

*C. elegans* NMY-2 is a homolog to the human NMHC2A (or MYH9, S7 Fig), and NMY-2 function has been linked to Src-mediated signaling [29]. Using CRISPR/Cas9 genome editing, we introduced point mutations at the conserved tyrosine 163 (Tyr[163]), corresponding to Tyr[158] in NMHC2A (S7 Fig), by replacing it with either a non-phosphorylatable (Y163F) or a phosphomimetic (Y163E) residue in the endogenous *nmy-2* locus.

We successfully generated a homozygous strain expressing NMY-2(Y163F), which showed normal brood size and embryonic viability comparable to the wild-type (Fig 5A). In contrast, we were unable to propagate *nmy-2(Y163E)* animals, as homozygous animals were sterile and therefore unable to lay embryos, suggesting that constitutive phosphorylation of Tyr[163] may impact gonad function. As the animals expressing NMY-2(Y163E) could not be propagated, we focused our analysis on *wild-type* and *nmy-2(Y163F)* animals.

We assessed whole-animal intoxication using dose–response survival assays across increasing concentrations of different PFTs, including LLO, PFO and a pore-deficient PFO (PFO inactive). Survival of *wild-type* animals was not

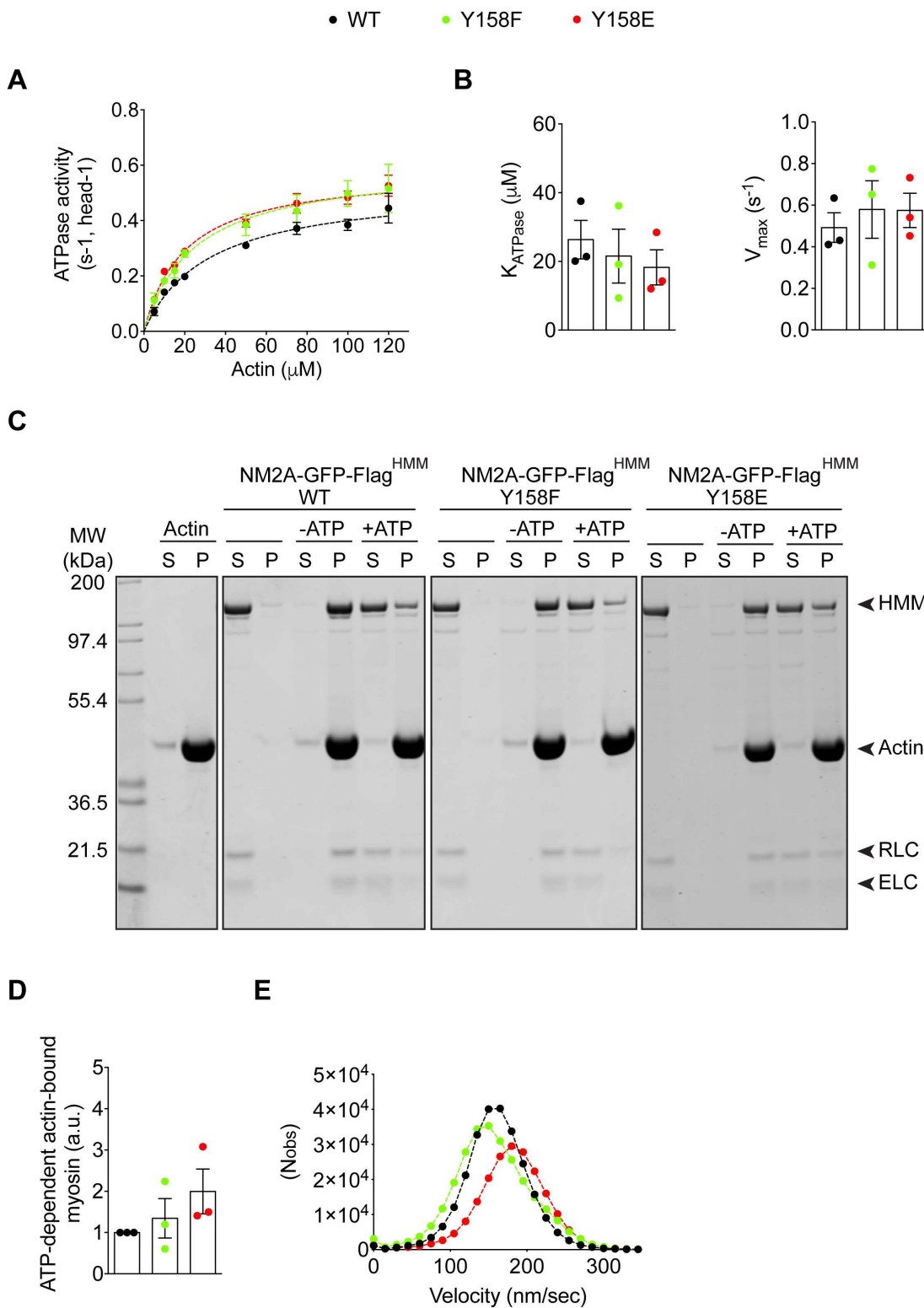

**Fig 4. Phosphorylation of NMHC2A in Tyr[158] does not affect NM2A mechanical or kinetic properties.** (A) Actin-activated Mg-ATPase activity of heavy meromyosin (HMM) NM2A-GFP-Flag[HMM] -WT, -Y158F and -Y158E determined by the conversion of NADH into NAD[+]. Three independent preparations of each NM2A-GFP-Flag[HMM] variant were used in at least 3 independent experiments. Data sets were fitted to a hyperbolic equation to determine

the kinetic constants, $K_{ATPase}$ and $V_{max}$. (B) Plot of the $K_{ATPase}$ and $V_{max}$ values. Data represent the mean±SEM of at least 6 independent experiments (per actin concentration) and using the 3 different preparations of NM2A-GFP-Flag[HMM] variants. Each dot represents the value obtained for a single preparation. Values are the mean±SEM ($n=3$). (C) Representative actin co-sedimentation assay of NM2A-GFP-Flag[HMM] WT, -Y158F and -Y158E. Supernatants (S) and pellets (P) obtained from ultracentrifugation of mixtures of either NM2A-GFP-Flag[HMM]-WT, -Y158F or -Y158E with or without 10 µM actin and 1mM ATP, are shown by Coomassie blue staining. Actin alone is also shown. Regulatory light chain (RLC); Essential light chain (ELC). (D) Quantification of co-sedimentation assays showing myosin fraction bound to actin (pellet) in the presence of ATP. Each dot corresponds to a single preparation of each NM2A-GFP-Flag[HMM] variant. Values are the mean±SEM ($n=3$). (E) Gaussian distribution of the velocity of actin filaments moving on top of either NM2A-GFP-Flag[HMM]-WT, -Y158F or -Y158E. Values were obtained from at least 5 independent motility assays (S2 Movie) performed on 3 independent NM2A-GFP-Flag[HMM] preparations. 500 to 1300 filaments were quantified for each condition and per independent experiment.

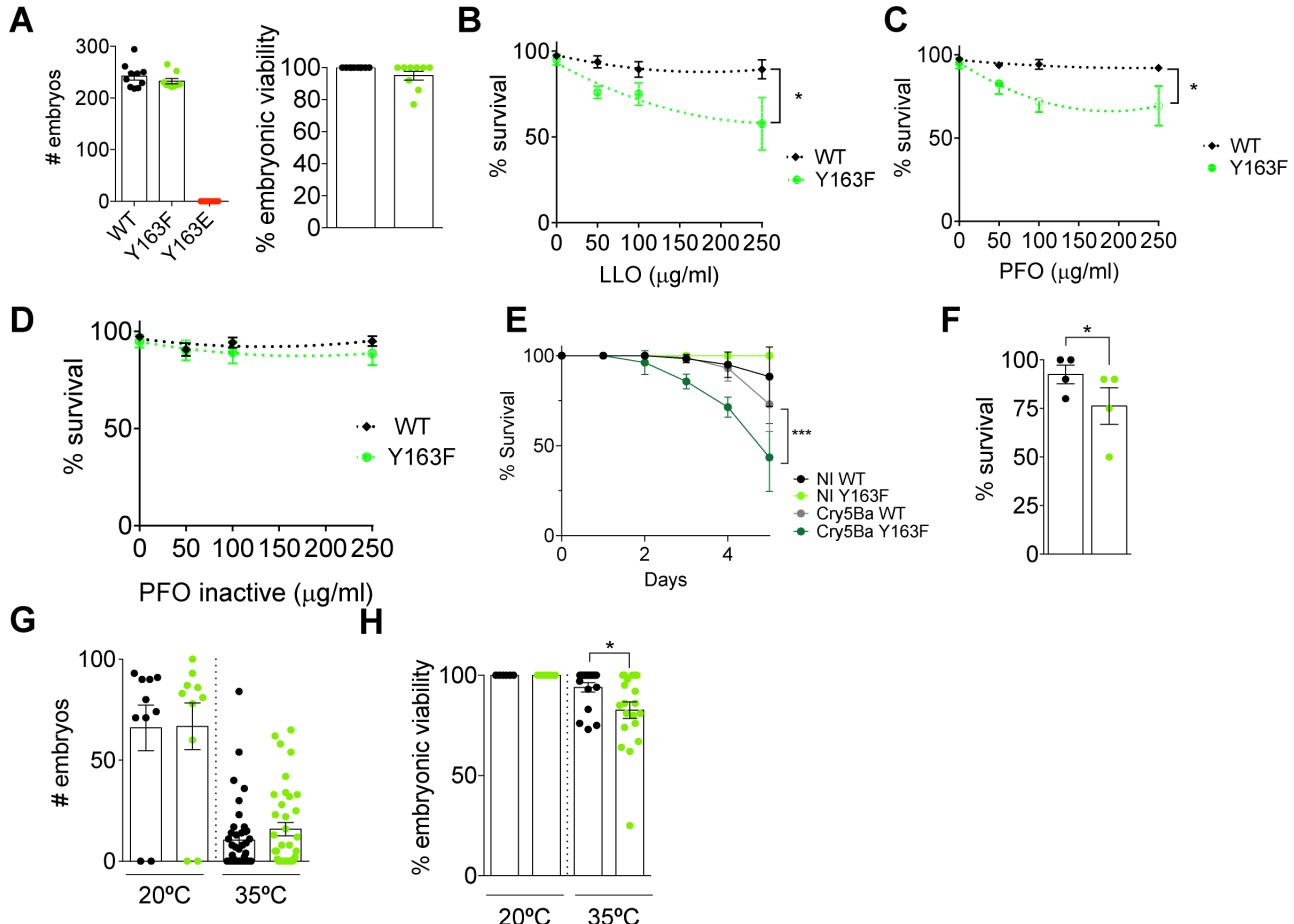

**Fig 5. Regulation of NMY2 pTyr[163] is necessary for the survival of *C. elegans* to intoxication and heat stress.** (A) Mean brood size and embryonic viability±SEM in *C. elegans* expressing wild-type NMY-2(WT), NMY-2(Y163F) or NMY-2(Y163E). On the left, each dot represents the brood size of a single animal. On the right, each dot represents the embryonic viability (in %) of embryos laid by each animal. (B-D) Viability scores based on movement of *wild-type* (WT) and *nmy-2(Y163F)* animals after 5-day exposure to growing concentrations of (B) LLO (C) PFO and (D) PFO inactive. Data are from three independent experiments performed in triplicate. *p*-values were calculated using two-way ANOVA with Sidak's *post hoc* analysis, *$p<0.05$. (E) Mean percentage of survival±SD of *wild-type* (WT) and *nmy-2(Y163F)* animals fed for 5 days with control *E. coli* (NI WT; NI Y163F) or with *E. coli* expressing the Cry5Ba toxin (Cry5Ba WT; Cry5Ba Y163F). Each curve represents the mean of 3 independent experiments with 30 animals per condition. *p*-values were calculated using two-way ANOVA with Sidak's *post hoc* analysis, ***$p<0.001$. (F) Mean percentage of survival±SEM of *wild-type* and *nmy-2(Y163F)* animals 24 h post a 1 h heat shock at 35 °C. Each dot represents one independent experiment with 8-10 animals per condition. *p*-values were calculated using two-tailed paired Student's *t*-test, *$p<0.05$. (G) Mean brood size±SEM of the animals that survived the 1 h heat shock at 35 °C. Each dot represents the brood size of a single animal. (H) Mean percentage of viability±SEM of the embryos laid by the animals that survived the 1 h heat shock at 35 °C. Each dot represents the percentage of viable embryos laid by a single animal. *p*-values were calculated using two-tailed paired Student's *t*-test, *$p<0.05$.

significantly affected by any of the toxins at the concentrations tested (Fig 5B-5D). In contrast, *nmy-2(Y163F)* animals showed a dose-dependent reduction in survival in response to increasing concentrations of both LLO and PFO (Fig 5B and 5C). Importantly, the inactive PFO variant did not cause significant toxicity in either *wild-type* or *nmy-2(Y163F)* animals (Fig 5D), indicating that the observed effects depend on pore-forming activity. Together, these data support a conserved functional role for NMHC2A/NMY-2 tyrosine phosphorylation in organismal responses to PFTs intoxication.

*C. elegans* is naturally susceptible to infection by *Bacillus thuringiensis*, which produces the PFT Cry5Ba and reduces nematode viability [30]. This infection system is a well-established *in vivo* model for dissecting host defense pathways against PFTs [31–34]. Motivated by the strong track record of this model, we examined the susceptibility of *nmy-2(Y163F)* mutant animals to Cry5Ba. *Wild-type* and *nmy-2(Y163F)* animals were fed with control or Cry5Ba-expressing *E. coli* starting with L4 larvae and the number of survivors was quantified daily, over a period of 5 days. While *wild-type* animals fed with control *E. coli* (NI WT) survived and appeared healthy over time, those fed with Cry5Ba-expressing *E. coli* (Cry5Ba WT) showed decreased survival (Fig 5E). Similarly, *nmy-2(Y163F)* animals showed normal survival when fed with control *E. coli*, but were significantly more susceptible to Cry5Ba-expressing *E. coli* than those expressing wild-type NMY-2 (Fig 5E). Our data suggest that proper regulation of NMY-2 phosphorylation at Tyr163 contributes to mounting an effective response to Cry5Ba intoxication in this system.

To address the possibility that NMY-2 phosphorylation at Tyr163 might also be important for the response to other sources of stress, we assessed its role in heat shock conditions. Testing heat shock stress was a logical choice, as our previous work demonstrated that PFTs target the endoplasmic reticulum (ER) and affect heat shock proteins, including the ER-resident chaperone Gp96 [4]. L4 larvae were subjected to heat shock for 1 h at 35 ºC and worm survival was assessed 24 h later. *nmy-2(Y163F)* animals showed increased susceptibility to heat shock (Fig 5F). Moreover, the number of embryos laid by heat-shocked *wild-type* and *nmy-2(Y163F)* animals was similar (Fig 5G), but embryo viability was lower for heat-shocked *nmy-2(Y163F)* progenitors (Fig 5H).

Altogether, these data demonstrate that in *C. elegans* the regulation of NMY-2 Tyr163 phosphorylation is important for survival responses to heat shock and PFT intoxication, suggesting a broad involvement in response to different stresses.

## Discussion

Our study uncovers a previously unrecognized host protective pathway activated in response to bacterial PFTs. We demonstrate that LLO is both necessary and sufficient to trigger Src activation and to induce the specific phosphorylation of NMHC2A at Tyr158. We further show that this response is not restricted to LLO, as related bacterial PFTs, including PLY and PFO, similarly induce NMHC2A tyrosine phosphorylation in a pore-dependent manner, thus establishing a conserved mechanism responding to pore formation and the consequent influx of calcium. We reveal that tyrosine phosphorylation of NMHC2A contributes to membrane damage response independently from affecting its canonical motor activity, highlighting its role in cellular stress responses. Finally, the enhanced sensitivity to PFT intoxication of *C. elegans* carrying a mutation in the homologous phosphorylation site of NMHC2A suggests evolutionary conservation of this NMHC2A-dependent protective pathway. Together, these findings define a novel mechanism whereby PFTs activate NMHC2A phosphorylation to promote cytoskeletal remodeling and respond to membrane damage.

LLO activates host cell signaling through both pore-dependent and pore-independent mechanisms [8,9,35]. Calcium influx through LLO pores leads to long-lasting oscillations of intracellular calcium concentrations [36], which initiate several signaling cascades [6]. In parallel, pore-independent mechanisms involve lipid raft clustering triggered by LLO-binding to cholesterol at the cell membrane, promoting the recruitment and activation of tyrosine kinases, particularly Src and Src-family kinases [8,9,35]. Here, we show that LLO exposure leads to rapid and robust Src activation and accumulation at specific PM sites, alongside cortical accumulation of NMHC2A phosphorylated in Tyr158. Importantly, these NMHC2A-enriched cortical sites correspond to sites of PM damage [4], supporting a model in which Src activation

is spatially confined to membrane injury zones. This newly identified signaling pathway is triggered by pore formation and calcium entry, although it does not exclude additional signaling initiated by LLO binding to membrane cholesterol. The spatial confinement of Src activity and NMHC2A Tyr[158] phosphorylation likely contributes to the immediate cellular response to PM damage, potentially facilitating actomyosin-dependent responses to damage and localizing signaling to the injury sites. Consistent with this interpretation, we find that this signaling pathway is broadly activated in response to other PFTs of the same family, suggesting a conserved host response to PFT-induced PM damage. Moreover, this pathway is activated during *L.monocytogenes* infection, and our data provide evidence that it is mediated by LLO. Notably, Src activity is also essential for *L. monocytogenes* entry into host cells [19], revealing a dual role for Src signaling in infection: facilitating bacterial invasion while initiating host responses to PM damage. This duality highlights the complexity of host-pathogen interactions, where a single host kinase can support pathogen invasion and trigger host defense responses. Finally, while our data support a pore- and calcium-dependent mechanism, the possibility remains that NMHC2A tyrosine phosphorylation reflects a broader cellular stress or innate immune response, rather than pore formation alone, underscoring the multifaceted nature of host signaling pathways engaged during bacterial infection.

Cortical tension arises from the contractile activity of the actomyosin cytoskeleton, generated by NMII activity on actin filaments beneath the PM [37]. Gradients in cortical tension lead to cell shape deformations required for cell migration, division and morphogenesis [37,38], all processes driven by NMII activity [39,40]. Src kinase has been involved in NMII-dependent processes, including cell adhesion and migration [24,41,42] and establishment of cell polarity [41,42]. Of particular interest here, Src activity was shown to regulate PM blebbing upon osmotic and non-osmotic stress by triggering MLCK- and/or ROCK-dependent pathways that ultimately regulate NMII activity [43–47]. PM blebs induced by PFTs are extruded or retracted in a process mediated by NM2A and are associated with NMHC2A cortical accumulation [4,10,16,48]. Importantly, we show here that inhibition of Src activity does not prevent the formation of these accumulations but alters their spatial distribution, suggesting that Src is not essential for NMHC2A recruitment *per se*, but coordinates its remodeling and reorganization. This coordination is functionally relevant, as Src-deficient cells are more susceptible to LLO-induced PM damage. Our data position Src kinase as a key modulator of cortical tension during the host response to PM injury, acting via phosphorylation of NMHC2A at Tyr[158], a modification that reorganizes cortical actomyosin architecture and contributes to PM repair. We cannot exclude the contribution of additional NMHC2A phosphorylation events, including serine and alternative tyrosine residues, to this response. Notably, Src kinases have been shown to phosphorylate NMHC2A at Tyr[277] and Tyr[1805], driving its accumulation at the PM during viral entry [49]. Together, our findings reinforce the existence of a Src–NMII signaling axis that modulates cortical tension in response to extracellular stimuli.

NMHC2A contains the conserved NM2A motor domain, where the ATP and actin-binding sites are located [11,50]. Although Tyr[158] is located in the NMHC2A motor region in the proximity of the ATPase domain [51], we showed through biochemical assays that amino acid substitutions mimicking permanent phosphorylation or absence of it (Y158E, Y158F, respectively) do not directly impact NMHC2A ATPase activity, actin binding, or actin filament translocation. These findings suggest that the phosphorylation of Tyr[158] does not regulate the motor function of NM2A, but likely serves as a signaling module, possibly influencing the assembly of NMII filaments or their interaction with other partners [30]. We propose that this modification contributes to the spatiotemporal organization of actomyosin networks, including the formation of NMII stacks or heterotypic filaments with specialized functions [13,30,52–55]. Overall, our data support a model in which NMHC2A functions as a phosphorylation-responsive sensor, translating toxic stress signals into cytoskeletal reorganization.

The physiological relevance of the phosphorylation of NMHC2A in Tyr[158] was further supported by *in vivo* experiments in *C. elegans*. The expression of a phosphomimetic form (Y163E) resulted in sterility, possibly reflecting a toxic effect or dominant-negative aggregation phenotype, as previously observed in human cells [24]. In contrast, the expression of the non-phosphorylatable mutant (Y163F) did not significantly affect gonad function but resulted in increased susceptibility to

the PFTs LLO, PFO and Cry5Ba, as well as to heat-shock stress, indicating that phosphorylation at this site is necessary for an effective response to multiple PFT challenges and for survival under stress conditions. Although our data indicate that, in epithelial cells, NMHC2A tyrosine phosphorylation depends on pore formation and calcium entry, it is also possible that this modification forms part of a broader stress response triggered by innate immune signaling pathways activated during PM damage. Several components of the damage response and innate immune system are known to converge on Src-family kinase activation [56], raising the possibility that NMHC2A phosphorylation may represent an integrated outcome of both pore-dependent and stress-induced signaling. Whether Src kinase is activated in *C. elegans* response to PFTs needs further investigation. These findings support a conserved functional role for NMHC2A/NMY-2 tyrosine phosphorylation in the organismal responses to environmental insults.

Interestingly, although a direct link between Src-family kinases and NMII has not been reported, previous studies in *C. elegans* suggest a possible functional connection. Src-mediated signaling regulates key developmental processes such as asymmetric cell division required for endoderm specification and cell migration during gonad morphogenesis [29,57,58]. During early embryogenesis, NMY-2 is required for Src-dependent phosphotyrosine accumulation at specific cell-cell interfaces [29,57], where NMY-2 also plays a role in the maintenance of cellular asymmetries and cell boundaries [29]. Although we do not demonstrate Src activation in *C. elegans* upon PFT exposure, nor show that Src phosphorylates NMY-2 in this system, our findings collectively suggest that a Src- NMHC2A/NMY-2 pathway may represent a conserved mechanism for regulating cortical actomyosin networks and coordinating responses to bacterial infection.

## Materials and methods

### Cell lines

HeLa cells (ATCC CCL-2) were grown in DMEM with glucose and L-glutamine (Lonza), supplemented with 10% fetal bovine serum (FBS; Biowest) and maintained at 37 °C in a 5% $CO_2$ atmosphere. Sf9 cells (ATCC CRL-1711) were grown in Grace's Insect Medium (Gibco) supplemented with 10% FBS (Biowest) and maintained at 27 °C in a humidified atmosphere without $CO_2$.

### Reagents, toxins, antibodies and dyes

Dasatinib (Santa Cruz Biotechnology) was used at 300 nM in complete medium for 1 h prior to LLO intoxication. LLO, PLY, PFO and PFO inactive were expressed in *E. coli* BL21(*DE3*) and purified as described in [59]. Intoxications and washes were carried out in Hanks' balanced salt solution (HBSS, Lonza). The following antibodies were used at 1/200 for immunofluorescence microscopy (IF) or 1/1,000 for immunoblotting (IB): rabbit anti-Src pTyr[419] (#44-660G, Thermo Fisher), mouse anti-active Src (#AHO0051, Thermo Fisher), rabbit anti-Src (#sc-18, Santa Cruz), mouse anti-β-actin (#A1978, Sigma); mouse anti-phosphotyrosine (#05–321, Millipore); rabbit anti-NMHC2A (#M8064, Sigma); rabbit anti-NMHC2A pY158 (#AP3775a, Abgent). For IF analysis, DNA was stained with 4′,6-diamidino-2-phenylindole dihydrochloride (DAPI; Sigma) and actin with Rhodamine Phalloidin (Thermo Fisher Scientific) at 1/200; secondary antibodies were used at 1/500: goat anti-rabbit Alexa Fluor 488 (Invitrogen), goat anti-rabbit Alexa Fluor 594 (Invitrogen), goat anti-mouse Cy3 (Jackson ImmunoResearch). For IB, goat anti-rabbit or anti-mouse HRP (PARIS) and donkey anti-rabbit or anti-mouse IRDye 680 or 800 (LI-COR Biosciences) were used at 1/10,000. For dye uptake assays using high-content microscopy, PhenoVue Draq7 (Perkin Elmer ILC) was used at 60 nM and Hoechst 33342 (Perkin Elmer ILC) was used at 100 ng/ml.

### Plasmids

Kras-Src FRET biosensor was a gift from Yingxiao Wang (Addgene plasmid # 78302) [22]. Plasmids allowing the expression of wild-type Src kinase (WT), constitutively active Src (CA) and kinase dead Src (KD) were kindly provided by S. J. Parsons (University of Virginia) [60]. pFastBac-NMHC2A[HMM]-GFP-Flag-WT, pFastBac-RLC and -ELC were described in [61]. Insertion of point mutations, Y158F and -Y158E, was achieved by site-directed mutagenesis using QuickChange II

Site-directed mutagenesis kit (Agilent) as described in [17]. pET28 His6-mEGFP-D4, allowing the expression and purification of PFO inactive, was a gift from Catherine Tomasetto (Addgene plasmid # 226372) [62].

## Infection experiments

Wild-type (EGDe, WT) and ΔLLO strains of *L. monocytogenes* were grown overnight at 37°C, with shaking, in brain–heart infusion (BHI; BD-Difco), sub-cultured at a 1:10 dilution until optical density of 0.6–0.8, washed in DMEM and inoculated into cells at a multiplicity of infection of 200.

## Immunoprecipitation

HeLa cells ($6 \times 10^6$ cells) treated as indicated or infected for 10 minutes or 1h, were washed with PBS 1x and lysed in RIPA buffer (Santa Cruz Biotechnology). Lysates were recovered after centrifugation at 15,000 x*g* (10 min at 4 °C). Cell lysates (approximately 500 µg) were pre-cleared with protein A-immobilized PureProteome magnetic beads (Millipore) and incubated overnight (4 °C) with 1.5 µg of anti-phosphotyrosine antibody (4G10 Millipore). Immune complexes were captured with 50 µl of protein G-immobilized magnetic beads (Millipore) and processed for immunoblotting.

## Immunoblot assays

Protein extracts of HeLa cells were recovered in sample buffer (0.25 mM Tris-Cl, pH 6.8; 10% SDS; 50% glycerol; and 5% β-mercaptoethanol), resolved by SDS–PAGE (10% acrylamide) and transferred onto nitrocellulose membranes using a TransBlot Turbo system (BioRad), at 0.3 A for 1h. Primary and secondary HRP/IRDye-conjugated antibodies were diluted in TBS-Tween 0.1% (150 mM NaCl; 50 mM Tris–HCl, pH 7.4; and 0.1% Tween) with 5% (m/v) milk. Washes were performed with TBS-Tween 0.2%. Signal was detected using Western Blotting Substrate (Thermo Fisher Scientific) and collected in a ChemiDoc XRS+ System with Image Lab Software (BioRad). Alternatively, the signal was detected using an Odyssey CLx Imager with Image Studio v5.2 Software (LI-COR Biosciences). For dot blots, samples were lysed in RIPA buffer (Santa Cruz Biotechnology) supplemented with protease and phosphatase inhibitors (Roche). Protein concentration for each sample was quantified using Bradford reagent (Biorad) and 10 and 20 µg of protein were spotted onto a nitrocellulose membrane. After complete drying for 30 min at room temperature, the membranes were blocked and primary and secondary HRP-conjugated antibodies were diluted in TBS-Tween 0.1% with 5% BSA (m/v). The signal was developed using Western Blotting Substrate (Thermo Fisher Scientific) and detected in Chemidoc Touch Gel Imaging System (BioRad).

## Immunofluorescence microscopy

Cells were fixed in 3% paraformaldehyde (15 min), quenched with 20 mM $NH_4Cl$ (1 h) and permeabilized with 0.1% Triton X-100 (5 min). Coverslips were incubated for 1 h with primary antibodies, washed three times in PBS 1x and incubated 45 min with secondary antibodies and when indicated, stained with Rhodamine Phalloidin and DAPI. Antibodies and dyes were diluted in PBS containing 1% BSA. Coverslips were mounted onto microscope slides with Aqua-Poly/Mount (Polysciences). Images were collected with a confocal laser-scanning microscope (Leica SP5 II or SP8) and processed using Fiji or Adobe Photoshop software. The percentage of cells with NMHC2A accumulations and the number of those accumulations per cell were blindly quantified in at least 200 cells per sample. Positive cells displayed at least a distinct cortical focus enriched in NMHC2A, each focus was quantified as a single NMHC2A accumulation. The intensity of the pY158 signal at cortical and sub-cortical regions was calculated using ImageJ software. Cell outlines were manually defined based on F-actin staining to generate total cell ROIs. A reduced ROI was produced by proportionally shrinking the original ROI using the *Enlarge* function. The PM-associated region was defined by subtracting the reduced ROI from the total cell ROI using the XOR operation, yielding a peripheral band of constant width in which pY158 integrated density was measured. Sub-cortical pY158 was quantified by further proportionally reducing and subtracting the cortical ROI, generating a second concentric band immediately beneath the PM.

## Dye Incorporation assay coupled to high-content microscopy

HeLa cells ($5 \times 10^4$) treated with DMSO or 300 nM Dasatinib were left non-intoxicated or were intoxicated with increasing concentrations of LLO (0.25 nM, 0.5 nM, and 1 nM diluted in HBSS with calcium and magnesium) in the presence of Hoechst (100 ng/ml) and Draq7 (60 nM) nuclear dyes. Fluorescence intensity was measured using a high-content widefield microscope (Operetta CLS Revvity) using a 20x water and 1.0 NA objective, capturing 17 fields per well (approximately 3000 cells per well), with a time step of 5 minutes for a total of 45 min. Data were analyzed using Harmony software (Revvity). Nuclei labeled by Hoechst (total cells) and by Draq7 (LLO-permeabilized cells) were automatically segmented. Draq7 uptake was expressed as the percentage of Draq7-positive nuclei relative to Hoechst-positive nuclei.

## Fluorescence resonance energy transfer (FRET)

HeLa cells were seeded into Ibitreat μ-dishes (Ibidi), transfected to express ECFP/YPet Kras-Src biosensor [22], maintained in HBSS or HBSS supplemented with LLO (2 nM) at 37 °C in 5% $CO_2$, and imaged using a confocal laser-scanning microscope (Leica SP8) equipped with a 40x 1.4 NA objective. Cells were imaged for 10 min prior LLO addition. Image acquisition under LLO-intoxicated conditions started 20 s after LLO addition. ECFP/YPet fluorescence data sets with 0.42 μm z-steps were acquired every 30 s. Fiji was used for image sequence analysis and video assembly. The ECFP/YPet ratio was quantified on the entire surface of at least 30 cells.

## Flow cytometry

For flow cytometry, $5 \times 10^5$ cells seeded in six-well plates 24 h before treatment, were intoxicated as indicated and washed twice in cold PBS 1x, trypsinized and resuspended in 0.5 ml of cold PBS 1x supplemented with 2% FBS. They were further incubated with 2 μg/ml of Propidium Iodide (PI, Sigma) 5 min before flow cytometry analysis. At least 30000 events per sample were analyzed in a FACS Calibur cytometer (BD Biosciences), and data were analyzed using FlowJo (TreeStar).

## NM2A-GFP-Flag^HMM-*WT, -Y158F and -Y158E expression in* Sf9 *cells*

NM2A-GFP-Flag^HMM variants were expressed in Sf9 insect cells, following the Bac-to-Bac Baculovirus Expression System (Life Technologies) as recommended. Briefly, Sf9 cells were co-transfected with the pFastBac-NMHC2A^HMM-GFP-Flag-WT, or -Y158F, or -Y158E, pFastBac-RLC and -ELC bacmids, and incubated at 27 °C for 96–120 h. Human RLC (NP_291024.1) and mouse ELC (NP_034990.1) were used [63]. Supernatants were collected to harvest the viral stock (P1). Sf9 cells in suspension were then infected with P1 (MOI = 1) to obtain a higher viral titer (P2) and incubated at 27 °C with shaking. The process was repeated and the viral titer P3 was used to co-infect Sf9 cells for further expression and purification of the HMMs. Sf9 P3-infected cells were harvested after 72 h and used for purification or stored at -80 °C.

## NM2A-GFP-Flag^HMM-WT, -Y158F and -Y158E purification

Sf9 infected-cell pellets were resuspended in extraction buffer (15 mM MOPS pH 7.3, 200 mM NaCl, 10 mM MgCl₂, 1 mM EGTA, 3 mM NaN₃, 1 mM DTT, 0.1 mM PMSF, 5 μg/ml leupeptin, 2 phosphatase inhibitors tablets) and lysed by sonication. The lysate was incubated with 1 mM ATP for 15 min, centrifuged at 19000 rpm for 20 min and incubated with pre-washed anti-Flag-resin (#A2220, Sigma) for 3 h at 4 °C. Flag-resin was washed with high ionic strength buffer (10 mM MOPS pH 7.3, 0.1 mM EGTA, 3 mM NaN₃, 0.5 M NaCl, 1 mM ATP, 5 mM MgCl₂ and 0.1 mM PMSF), loaded into a column (4 °C) and further washed with 3 column volumes of low ionic strength buffer (10 mM MOPS pH 7.3, 0.1 mM EGTA, 3 mM NaN₃ and 0.1 mM PMSF). Proteins were eluted with 2 column volumes of Flag-peptide buffer (0.5 mg/ml Flag peptide, 10 mM MOPS pH 7.3, 0.1 mM EGTA, 3 mM NaN₃, 100 mM NaCl and 0.1 mM PMSF). Eluted fractions were concentrated and dialyzed

overnight in the dialysis buffer (10 mM MOPS pH 7.3, 0.1 mM EGTA, 3 mM $NaN_3$, 0.5 M NaCl and 1 mM DTT) at 4 ºC. Aliquots of the purified proteins were quickly frozen in liquid nitrogen and stored at -80 ºC.

## Single molecule electron microscopy

For NM2A-GFP-Flag[HMM] molecules in the absence of actin, samples were diluted to 50 nM HMM in 10 mM MOPS pH 7.0, 50 mM NaCl, 2 mM $MgCl_2$, 0.1 mM EGTA. NM2A-GFP-Flag[HMM] molecules were mixed with F-actin to give final concentrations of 100 nM of HMM and 500 nM of F-actin. Samples were handled as previously described [64]. Briefly, 5 μl of the sample was applied to a carbon-coated copper grid (pre-exposed to UV light to produce a hydrophilic surface) and stained with 1% uranyl acetate. Data were recorded at room temperature on a JEOL 1200EX II microscope equipped with an AMT XR-60 CCD camera. The Fiji FFT bandpass filter (40 – 2 pixels, auto-scaled) was applied to images prior to making figures, in order to enhance the clarity of individual molecules.

## ATPase activity assay

Actin-activated $Mg^{2+}$-ATPase activities were determined by previously described NADH-linked assay [65,66]. Purified NM2A-GFP-Flag[HMM] variants were phosphorylated for at least 30 min at room temperature, in adequate buffer conditions (0.2 mM ATP, 0.2 mM $CaCl_2$, 10 μg/ml MLCK and 1 μM CaM). $Mg^{2+}$-ATPase activities were determined in 10 mM MOPS, 0.1 mM EGTA, 50 mM NaCl, 2mM $MgCl_2$, 3.4 mM ATP, growing actin concentrations (0–120 μM), 40 units/ml lactate dehydrogenase, 200 units/ml pyruvate kinase, 1 mM phosphoenolpyruvate, 0.2 mM NADH and non-phosphorylated or phosphorylated HMMs (0.1 – 0.2 μM). The conversion of NADH into $NAD^+$ was determined by measuring the $A_{340}$ ($\epsilon$ = 6220 $M^{-1\ cm-1}$) every second for 20 min. Data were corrected for the background $Mg^{2+}$-ATPase activity of actin alone. The Mg-ATPase activity of the different NM2A-GFP-Flag[HMM] variants at 0 μM actin concentration was further subtracted to each data point. To determine the kinetic constants $V_{max}$ and $K_{ATPase}$, the experimental data sets were fitted to the Michaelis-Menten mathematical equation using Prism 8.

## Co-sedimentation assay

2 μM of each NM2A-GFP-Flag[HMM] variant was incubated for 30 min at room temperature with 10 μM f-actin in the absence or presence of 1 mM ATP, in a buffer containing 10 mM MOPS, 0.1 mM EGTA, 100 mM NaCl, 2 mM $MgCl_2$, 3 mM $NaN_3$, and 1 mM DTT. Samples containing only the NM2A-GFP-Flag[HMM] variants or filamentous actin were used as sedimentation controls. Samples were centrifuged at 100,000 xg for 30 min at 4 ºC. Supernatant and pellet fractions were recovered and the loading buffer was added equivalently. Each sample (20 μl) was analyzed on a 4–12% Bis-Tris gel (Life Technologies), stained with Coomassie Blue and scanned on an Odyssey system (Li-Cor Biosciences).

## *In vitro* motility assay

I*n vitro* motility assays were performed as described [67]. Briefly, 60 μl of NM2A-GFP-Flag[HMM] molecules (0.2 mg/ml) in motility buffer (MB: 20 mM MOPS, pH 7.3, 0.1 mM EGTA, 2 mM $MgCl_2$) with 0.5 M NaCl were trapped in a flow cell. NM2A-GFP-Flag[HMM] molecules attached to the coverslip were further incubated with 50 μl MB with 0.5 M NaCl and 1 mg/ml BSA, washed in MB with 50 mM NaCl (LS buffer) and incubated for 4 min with 35 μl of 1 mM ATP, 0.2 mM $CaCl_2$, 1 mM ATP, 1 μM CaM, 1 nM MLCK and 10 μM of unlabeled actin in LS buffer. Coverslips were washed in LS buffer and incubated with 30 μl of labeled actin filaments (20 nM Rhodamine phalloidin actin in 50 mM DTT) for 35 s. The excess solution was removed and the flow cell was loaded for 1 min with 40 μl of MB with 0.7% methylcellulose, 1mM ATP, 50 mM KCl, 50 mM DTT, 2.5 mg/ml glucose, 2.5 μg/ml glucose oxidase, and 45 μg/ml catalase. The slides were imaged at 25 ºC, at 5 s intervals for 2 min, on an inverted Nikon Eclipse Ti-E microscope with an H-TIRF module attachment, a CFI60 Apochromat TIRF 100x Oil Immersion Objective Lens (N.A. 1.49, W.D. 0.12 mm, F.O.V 22 mm) and an EMCCD camera (Andor iXon Ultra 888

EMCCD, 1024 × 1024 array, pixel size: 13 µm). Velocity of the actin filaments on top of the different NM2A-GFP-Flag[HMM] molecules was quantified using the FAST algorithm described in [68].

## *C. elegans* strains

*C. elegans* strains were maintained at 20 ℃ on nematode growth medium (NGM) plates previously seeded with OP50 *E. coli*. Strains harboring NMY-2 point mutations (Y163F and Y163E) were generated by CRISPR/Cas9 (S1 Table). Briefly, the gonads of young adult *wild-type* animals (N2 strain) were injected with three different single guide RNA sequences (sgRNAs) and a single-stranded repair template (ssODN; IDT ultramer) to edit the *nmy-2* locus (details available in S2 Table). The sgRNAs were individually expressed from the pDD162 vector, which also contains the Cas-9 gene re-encoded for *C. elegans* [69]. The injection mix also contained a sgRNA and a single-stranded repair template carrying the R92C mutation on the *dpy-10* gene, to allow the identification of successfully injected worms through the described roller pheno-type [70,71]. To screen for animals carrying the targeted modifications, amplification of a *nmy-2* fragment was performed using specific primers (Diagnosis primers, available in the supplementary material), followed by digestion with the *Hind*III or *Eco47*III restriction enzymes that recognize the restriction site present in modified animals only. The successful editing of the *nmy-2* locus was confirmed by genomic DNA sequencing. *nmy-2* mutant strains were outcrossed six times with N2 animals to eliminate possible off-target mutations.

## Brood size and embryo viability in *C. elegans*

Synchronized L1 animals of N2 and GCP693 strains were grown in NGM plates seeded with OP-50 *E. coli* at 20 ℃ for 72 h (1-day adults). Adult animals were singled out onto fresh NGM plates and the total number of eggs they laid over 24 h was counted for brood size quantification. For quantification of embryonic viability, the percentage of embryos that hatched was determined 24 h after embryos were laid.

## Toxin soaking assay in *C. elegans*

Toxin exposure experiments were performed by incubating worms in liquid medium supplemented with defined concen-trations of toxins and food [72]. For each condition, ~ 25 L4 stage animals of N2 and GCP693 strains were transferred into individual wells of a 96-well plate (in triplicate). Each well contained a total volume of 200 µl of S-medium supplemented with OP-50 *E. coli* ($OD_{600}$ ~ 2–3) and 50 µM 5-fluoro-20-deoxyuridine (FUdR) to inhibit progeny production. Toxins (LLO, PFO or PFO inactive) were prepared in 20 mM HEPES (pH 8) at final concentrations of 50, 100 and 250 µg/ml. Plates were incubated at 25 ℃ in a humid chamber for 5 days. Animals were scored for viability by touch-provoked movement.

## Cry5Ba intoxication assay

Synchronized L1 animals of N2 and GCP693 strains were grown in OP-50 *E. coli*-seeded NGM plates at 20 ℃ for 50 h (L4 stage) and then transferred to NGM plates previously seeded with control (empty vector pQE) or Cry5Ba-expressing BL21(*DE3*) *E. coli* [72]. Adult animals were transferred into freshly-seeded plates every 2 days. Moving and dead animals were counted every 24 h, the ones not responsive to touch were considered dead.

Briefly, saturated cultures of *E. coli* transformed with the pQE empty vector or pQE-Cry5Ba vector were diluted (10x) in LB with 100 µg/ml ampicillin and grown for 1 h at 37 ℃ with shaking. Expression of Cry5Ba was induced with 1 mM IPTG for 3 h at 30 ℃. 100 µl of each culture at OD = 2 ± 0.1 were plated onto NGM plates containing 2 mM IPTG and 100 µg/ml ampicillin, and incubated at 25 ℃ overnight.

## Heat shock assay

Synchronized L1 animals of N2 and GCP693 strains were grown as described above for 50 h. L4 animals of each strain were transferred to fresh NGM plates and maintained at 20 ℃ or incubated in a water bath at 35 ℃ for 1 h. Plates were

then kept at 20 ºC for 24 h. Surviving animals were singled out on fresh NGM plates and allowed to lay eggs for 24 h at 20 ºC. Brood size and embryo viability were assessed 24 h later.

## Statistical analysis

Statistical analyses were carried out with Prism 8 (version 8.1.1, GraphPad Software, La Jolla California USA), using one-way ANOVA with *Dunnett's* *post hoc* analysis to compare different means in relation to a control sample and Tukey's *post hoc* analysis for pairwise comparisons of more than two different means. Two-way ANOVA with Šídák's *post hoc* analysis was used to compare each sample mean with another sample mean, in the same condition. Two-tailed unpaired and paired Student's *t*-test was used for comparison of means between two samples.

## Supporting information

**S1 Fig. *Listeria monocytogenes* infection of HeLa cells induces the tyrosine phosphorylation of NMHC2A and the activation of Src kinase in an LLO-dependent manner. (A, B)** Levels of NMHC2A measured by immunoblots on whole-cell lysates (WCL) and immunoprecipitated (IP) fractions of pTyr proteins (IP pTyr) from HeLa cells left non-infected (NI) and infected with L. monocytogenes wild-type (WT) or the isogenic strain lacking LLO expression (ΔLLO) for 1 hour. Actin was used as loading control. (B) Levels of NMHC2A in the IP pTyr fraction (NMHC2A pTyr) were quantified and normalized to those detected in the WCL (NMHC2A WCL). Each dot corresponds to an independent experiment. Data correspond to mean ± SEM ($n = 4$); *p*-value was calculated using a Kruskal Wallis test with Dunn's *post hoc* analysis, *$p < 0.05$. **(C-F)** Src kinase is activated by LLO in HeLa cells. **(C)** Immunoblot on total lysates of HeLa cells, non-intoxicated (NI) or intoxicated for 5 min with 0.5 nM LLO, showing the levels of active Src through detection of Src non-phosphorylated at Tyr530. Actin detection served as loading control. **(D)** Quantification of active Src (Tyr530 non-phospho Src) signals normalized to the actin levels. Each dot corresponds to a single independent experiment. Values are the mean ± SEM (n = 6); p-values were calculated using two-tailed unpaired Student's t-test, *p < 0.05. (E) Immunoblot on total lysates of HeLa cells, non-infected (NI) or infected for 10 minutes with L. monocytogenes wild-type (WT) or the isogenic strain lacking LLO expression (ΔLLO), showing the levels of active Src through detection of Src non-phosphorylated at Tyr530 and of total Src (Src). Actin detection served as loading control. (F) Quantification of active Src (Tyr530 non-phospho Src) signals normalized to the Src total levels. Each dot corresponds to a single independent experiment. Values are the mean ± SEM (n = 4); *p*-values were calculated using a Kruskal Wallis test with Dunn's *post hoc* analysis, *$p < 0.05$.
(TIF)

**S2 Fig. NMHC2A tyrosine phosphorylation and Src activation is a common cellular response to different bacterial PFTs belonging to the same family as LLO. (A-D)** Levels of NMHC2A measured by immunoblots on whole-cell lysates (WCL) and immunoprecipitated (IP) fractions of pTyr proteins (IP pTyr) from HeLa cells left non-intoxicated (NI) or intoxicated with LLO (0.5 nM, 10 min) in the presence (LLO) or absence of extracellular calcium (LLO Ca$^{2+}$-free) or with PFO (0.5 nM, 10 min) or with PFO lacking ability to form pores (PFO inactive, 5 nM, 10 min) or with PLY (0.25 nM, 10 min). Actin was used as loading control. **(B, D)** Levels of NMHC2A in the IP pTyr fraction (NMHC2A pTyr) were quantified and normalized to those detected in the WCL (NMHC2A WCL). Each dot corresponds to an independent experiment. Data correspond to mean ± SEM ($n = 4$, $n = 3$ for PFO, $n = 2$ for PLY); *p*-value was calculated using a Tukey's *post hoc* analysis, *$p < 0.05$ and **$p < 0.01$. **(E-G)** Src kinase is activated by PLY in HeLa cells. **(E)** Immunoblot on total lysates of HeLa cells, non-intoxicated (NI) or intoxicated with PLY (0.25 nM, 5 min), showing the levels of Tyr419-phosphorylated Src (Src pTyr419) and total Src. Actin was used as loading control. **(F)** Quantification of Src pTyr419 signals normalized to the levels of total Src. Each dot corresponds to an independent experiment. Values are the mean ± SEM ($n = 3$). **(G)** Confocal microscopy images of NI or PLY-intoxicated (0.25 nM, 10 min) HeLa cells,

immunolabeled for active Src (red) and NMHC2A (green). Insets show PLY-induced cortical accumulations of NMHC2A enriched in active Src. Scale bar, 10 μm.
(TIF)

**S3 Fig. Src kinase inhibition increases permeability of HeLa cells upon LLO-intoxication. (A)** Confocal microscopy images of non-intoxicated HeLa cells in control (Mock and shCtr) and Src-impaired (Dasatinib-treated and shSrc) conditions. Cells were immunolabeled for NMHC2A (greyscale) and stained with DAPI (blue). Scale bar, 10 μm. **(B)** Percentage of Draq7 uptake in non-intoxicated (NI) or LLO-intoxicated (0.25 nM, 0.5 nM, or 1 nM) HeLa cells in control (black) and Dasatinib-treated (blue) conditions, measured over time (every 5 min, for 45 min) by high-content microscopy. Values are the mean ± SEM ($n = 2–3$); $p$-values were calculated using two-way ANOVA with Tukey's *post hoc* analysis, *$p < 0.05$, **$p < 0.01$, ***$p < 0.001$. **(C)** Average number of nuclei per well detected in each condition of (B) at 45 min. $p$-values were calculated using Kruskal Wallis test with Dunn's *post hoc* analysis, n.s. non-statistically significant. **(D)** Representative high-content microscopy images of Hoechst-positive nuclei (green) and Draq7-positive nuclei (red) in each condition of (B) at 0 min and 45 min. A random field was selected. Scale bar, 100 μm.
(TIF)

**S4 Fig. Representative flow cytometry gating strategy used to define the PI positive cells under non-intoxicated or LLO-intoxicated conditions in control and Src-impaired cells. (A)** Flow cytometry plots of ShCtr and ShSrc HeLa cells left non-intoxicated (nt) or intoxicated with LLO (llo). Cells were gated on forward (FSC) versus side scatter (SSC) to select the cell population (top). Next, all the subpopulations were analysed on the PI scatter (FL3) considering the unstained controls (unsdt) to establish PI-positive cells. **(B)** Gating strategy applied to cells treated with 0.05% Triton X-100 for 5 min to induce complete membrane permeabilization, used as a positive control for PI staining and to define the PI-positive population.
(TIF)

**S5 Fig. NMHC2A Tyr[158] localization and conservation throughout evolution. (A)** Ribbon representation of the head domain of NMHC2A (green) to show Tyr[158] (purple) and ADP (magenta) localization. The ELC (light yellow) and the actin-binding pocket (violet) are shown. PDB entry 1BR4. **(B)** NMHC2A amino-acid sequence analysis from different species and focused on the region involving the Tyr[158], adapted from (17).
(TIF)

**S6 Fig. Purified NM2A-GFP-Flag[HMM] molecules from baculovirus-infected Sf9 cells. (A)** SDS-polyacrylamide gel stained with Coomassie Blue showing the separation of the purified heavy mero-myosins (HMM), and the regulatory and essential light chains (RLC and ELC, respectively) for each NM2A-GFP-Flag[HMM] variant. **(B)** Representative single molecule electron microscopy images showing either a NM2A-GFP-Flag[HMM]-WT, Y158F or Y158E molecule. Scale bar, 10 nm. **(C)** Electron microscopy images showing the bound (arrow heads) and unbound (arrows) status of all purified NM2A-GFP-Flag[HMM] variants to actin filaments in the absence or presence of ATP. Scale bar, 100 nm.
(TIF)

**S7 Fig. Comparison of the amino acid sequence of the heavy chain of myosin 2A from different species.** Protein sequence analysis of the NMHC2A from *H. sapiens, M. musculus* and *C. elegans*, focused in the region containing the Tyr[158] in human NMHC2A. The position of the corresponding residue in the other organisms is indicated.
(TIF)

**S1 Table. *C. elegans* strains used in this study.**
(DOCX)

**S2 Table. shRNA sequences for Src knockdown in HeLa cells and single-guide RNAs, single-stranded repair templates and RNAi sequences used in *C. elegans*.** Highlighted in grey are the restriction sites used to screen the mutant animals. HindIII for Y163F and Eco47III for Y163E.
(DOCX)

**S1 Movie. Src is activated in response to LLO intoxication in HeLa cells.** HeLa cells expressing ECFP/YPet-based Src biosensor were imaged by time-lapse microscopy during 10 min (n on-intoxicated). LLO was added to a final concentration of 2 nM and cells were further imaged for 30 min. Sequential frames acquired every 30 s (15 frames per second display rate) show the ECFP (cyan) and YPet (yellow) fluorescence intensity separately, and the ECFP/YPet ratio. Scale bar, 10 μm.
(MOV)

**S2 Movie. NMHC2A pTyr does not affect the associated actin motility assay.** Rhodamine-phalloidin-labeled actin filaments moving on top of either the NM2A-GFP-Flag<sup>HMM</sup> -WT, -Y158F or -Y158E. Sequential frames were acquired every 1 s for 2 min (15 frames per second display rate). Scale bar, 20 μm.
(AVI)

## Acknowledgments

We are grateful to the members of i3S scientific facilities (ALM and Tracy) for technical assistance with microscopy and flow cytometry analysis, respectively. We acknowledge the NHLBI Electron Microscopy Core Facility for the assistance during TEM samples processing. We thank Raffi Aroian (UMass Medical School) for sharing plasmids encoding Cry5Ba and respective control, Charles Bond (University of Pennsylvania) for helping with the analysis of motility assays and Renato Socodato (Glial Cell Biology, i3S) for initial assistance with FRET experiments.

## Author contributions

**Conceptualization:** Cláudia Brito, Francisco S. Mesquita, Didier Cabanes, Sandra Sousa.

**Funding acquisition:** James R. Sellers, Didier Cabanes, Ana X. Carvalho, Sandra Sousa.

**Investigation:** Cláudia Brito, Francisco S. Mesquita, Joana M. Pereira, Daniel S. Osório, Neil Billington, Ricardo R. Lima, Sílvia Vale-Costa.

**Writing – original draft:** Cláudia Brito, Sandra Sousa.

**Writing – review & editing:** Cláudia Brito, Francisco S. Mesquita, Joana M. Pereira, Daniel S. Osório, Neil Billington, Sílvia Vale-Costa, James R. Sellers, Didier Cabanes, Ana X. Carvalho, Sandra Sousa.

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
