## [Decision Letter · Decision Letter 0]

31 Aug 2025

Src-dependent tyrosine-phosphorylation of NM2A has a protective role against bacterial pore-forming toxins

PLOS Pathogens

Dear Dr. Sousa,

Thank you for submitting your manuscript to PLOS Pathogens. After careful consideration, we feel that it has merit but does not fully meet PLOS Pathogens's publication criteria as it currently stands. Therefore, we invite you to submit a revised version of the manuscript that addresses the points raised during the review process.

Please submit your revised manuscript within 60 days Oct 30 2025 11:59PM. If you will need more time than this to complete your revisions, please reply to this message or contact the journal office at plospathogens@plos.org. Please include the following items when submitting your revised manuscript:

We look forward to receiving your revised manuscript.

Kind regards,

Francis Alonzo

Academic Editor

PLOS Pathogens

Michael Otto

Section Editor

PLOS Pathogens

Sumita Bhaduri-McIntosh

Editor-in-Chief

PLOS Pathogens

orcid.org/0000-0003-2946-9497

Michael Malim

PLOS Pathogens

orcid.org/0000-0002-7699-2064

**Additional Editor Comments:**

Three reviewers provided a thorough assessment of this manuscript. All felt that the work is potentially significant, drawing new insights on cell signaling in response to pore-forming toxin exposure. However, there was a general consensus that additional experimental support and/or relevant editing is needed to draw the stated conclusions. Reviewer 1 points out the importance of a more rigorous assessment of the relevance of pore formation to the noted phenotypes, clearer investigation of whether membrane repair is the outcome, and a deeper assessment of point mutants. Reviewer 2 points out areas for improved clarity in several areas and more thorough consideration of studies to better probe the model. Reviewer 3 echoes points made by reviewer 1, including use of a WT versus Δhly strain of Listeria and potentially inclusion other pore forming toxins (PLY/SLO) to bolster the relevance of the conclusions. Please see the detailed reviewer comments.

**Journal Requirements:**

At this stage, the following Authors/Authors require contributions: Cláudia Brito, Francisco Sarmento Mesquita, Joana Maria Pereira, Daniel S. Osório, Neil Billington, James R. Sellers, Didier Cabanes, Ana X. Carvalho, and Sandra Sousa. Please ensure that the full contributions of each author are acknowledged in the "Add/Edit/Remove Authors" section of our submission form.

- ® on page: 15

- TM on pages: 13, 14, and 16.

3) We notice that your supplementary Figures are included in the manuscript file. Please remove them and upload them with the file type 'Supporting Information'. Please ensure that each Supporting Information file has a legend listed in the manuscript after the references list.

Potential Copyright Issues:

i) Figure 2i. Please confirm whether you drew the images / clip-art within the figure panels by hand. If you did not draw the images, please provide (a) a link to the source of the images or icons and their license / terms of use; or (b) written permission from the copyright holder to publish the images or icons under our CC BY 4.0 license. Alternatively, you may replace the images with open source alternatives. See these open source resources you may use to replace images / clip-art:

5) Please provide a complete Data Availability Statement in the submission form, ensuring you include all necessary access information or a reason for why you are unable to make your data freely accessible. If your research concerns only data provided within your submission, please write "All data are in the manuscript and/or supporting information files" as your Data Availability Statement.

7)  Please ensure that the funders and grant numbers match between the Financial Disclosure field and the Funding Information tab in your submission form. Note that the funders must be provided in the same order in both places as well.

8) Please ensure that the affiliations of the authors listed on the manuscript title page (CITY) do exactly match with the affiliations provided in the online submission form

NOTE: Affiliations should include a department (if applicable), an institution, a CITY, and a country

**Reviewers' Comments:**

Reviewer's Responses to Questions

**Part I - Summary**

Reviewer #1: In this study the authors analyzed the cellular protective response to pore-forming toxins (PFTs), focusing on the role of non-muscle myosin II heavy chain A (NMHC2A) phosphorylation downstream of Src kinase activation. The authors report that Listeriolysin O (LLO) from Listeria monocytogenes induces Src-dependent phosphorylation of NMHC2A at tyrosine 158, which is required for cytoskeletal reorganization and plasma membrane (PM) repair but does not alter NMHC2A’s motor activity. They further demonstrate evolutionary conservation of this mechanism in Caenorhabditis elegans following exposure to Cry5Ba.

The manuscript addresses an important and timely question regarding how host cells counteract the membrane damage caused by PFTs. However, while the findings are potentially significant, several issues limit the current impact of the work and require further clarification.

1. The novelty of the work is not sufficiently highlighted. LLO-induced Src activation and NMHC2A phosphorylation during bacterial infection have both been reported (Sousa et al., 2007; Almeida et al., 2015). The authors should explicitly state what is new here: Is it the identification of Tyr158 as the relevant residue? The functional dissection separating NMHC2A’s motor activity from its role in membrane repair? Or the demonstration of evolutionary conservation using Cry5Ba?

2.The cross-species experiment with C. elegans and Cry5Ba is potentially interesting, but its mechanistic relevance remains unclear. How directly comparable are these systems? Does Cry5Ba also activate Src or a functional homolog in nematodes?, and does it target NMHC2A orthologs in a similar manner? The evolutionary conservation claim should be either substantiated with mechanistic data or substantially tone down.

Reviewer #2: In “Src-dependent tyrosine-phosphorylation of NM2A has a protective role against bacterial pore-forming toxins,” Brito et al identify a host protective pathway triggered by the addition of purified exogenous LLO where tyrosine kinase Src is activated, leading to phosphorylation of NMHC2A at tyrosine 158, that is critical for cytoskeletal reorganization and PM repair. The authors also use C. elegans to shown that phosphorylation of a NMHC2A homolog at the conserved tyrosine is required for survival under heat shock and after the organisms consume E. coli expressing a pore-forming toxin. This work shows a link between host sensing of PM damage and actomyosin remodeling.

Reviewer #3: In this manuscript by Brito, Mesquita, Pereira et al., the authors propose a host signaling pathway – through Src kinase and non-muscle heavy chain myosin 2A (NMHC2A) – that mobilizes cytoskeletal remodeling in the host to maintain plasma membrane integrity in response to damage from pore-forming toxins.

The authors first provide thorough evidence that NMHC2A and Src are both post-translationally modified after exposure to the pore-forming toxin, LLO. This results in the activation and re-localization of Src to the periphery of the cell – corresponding to where LLO-induced damages would be incurred, although importantly the authors never directly localize Src signaling to sites of membrane damage. Next, the authors demonstrate that Src is important for NMHC2A phosphorylation using various strategies spanning from the use of chemical inhibitors, knockdown of Src, as well as the use of constitutively active and kinase dead variants that cause dramatic phenotypes regarding the phosphorylation status of NMHC2A. The diversity of approaches used by the authors as well as the effect sizes of the quantifications presented (with exception of Figure 2J, please see below) nicely strengthen the claims made in the first two figures. The authors also provide thorough biochemical data using purified variants of phosphomimetic and phosphoablative point mutations to support the claim that the phosphorylation status of NMHC2A does not affect its kinetic properties with regards to ATPase activity nor its ability to associate with co-sedimentation with actin. Finally, the authors finish their manuscript with data supporting their claim that the phosphorylation status of the homologous protein in C. elegans is necessary for survival against the Cry5B pore-forming toxin and heat stress and extrapolate this finding to suggest that this is due to defects in membrane repair although membrane repair is never directly addressed in this model.

Despite strong validation of Src-NMHC2A signaling in the presence of LLO, multiple of the major conclusions of this manuscript are incompletely supported. One major conclusion is that pore forming toxins trigger this host cell response. This group had previously demonstrated that Listeria and E. coli infection (no pore forming toxin) can both trigger the response but the investigators never determined if Listeria’s induction was dependent on expression of LLO using an hly mutant. Similarly, while the investigators used purified LLO in this study, they did not have a pore formation deficient mutant and did not demonstrate that the signaling cascade is a broadly conserved response using another purified toxin in their robust in vitro signaling assays. Finally, the claim that this signaling facilitates membrane repair is not supported as the authors never determine if defects in signaling result in host cell death due to inability to repair the pores induced by LLO but rather they simply measure pore formation by PI which should not be different based on their hypothesis but rather the repair of these pores should be. Is it possible that disruptions in these pathways result in altered cholesterol concentrations leading to differential activity of LLO? Additionally, in this assay the shCtr and the Dasa mock are significantly different from one another even though they should be equivalent controls, thus making it difficult to believe that the differences in PI staining in the experimental samples are indeed biologically different.

Another major conclusion is that the phosphorylation does not affect the biochemical activity but does affect the response to LLO, however given that the point mutants utilized in the biochemical assays are never expressed in the cell culture models it is impossible to know if these mutations actually affect function in the cells either. It is possible that they affect protein protein interactions in their functional molecular complexes in the cell and this is not recapitulated in the in vitro biochemical system, but as they are never tested in the cell the impact of the biochemical characterization is unclear.

In sum, the experiments presented in this manuscript do a good job of dissecting this signaling pathway, demonstrating the biochemical impact of specific point mutations in NMHC2A and the impact of these mutations in a C. elegans model. However, the authors fall short of demonstrating that it is pore formation that triggers these responses, that the outcome of the response is membrane repair to protect from pore forming toxins and that the phosphomimetic and ablative mutations they studied biochemically have phenotypes in cellulo. Specific recommendations to address these concerns are:

**Part II – Major Issues: Key Experiments Required for Acceptance**

Reviewer #1: 1. Mechanistic depth and spatial dynamics

The central claim is that NMHC2A phosphorylation at Tyr158 is essential for cytoskeletal remodeling and PM protection. However, the study provides limited insight into how this modification coordinates actomyosin activity at specific sites of toxin-induced damage. Does phosphorylation regulate NMHC2A recruitment, assembly, or interaction with accessory proteins at PM lesions? Without this spatial and mechanistic resolution, the functional model remains incomplete.

2. Focus on a single phosphorylation site

By restricting analysis to Tyr158, the authors may be overlooking additional phosphorylation events or post-translational modifications that could also regulate NMHC2A. Given the complexity of Src signaling and cytoskeletal dynamics, the exclusive focus on one site appears narrow. At minimum, a broader phosphoproteomic analysis should be discussed, and ideally preliminary evidence provided.

Reviewer #2: This manuscript is clearly written and straight forward, however it is unclear whether activation of NM2A though Src is a mechanism that occurs in response to a true bacterial infection. The authors expose cell to exogenous LLO from a facultative intracellular pathogen. Typically, LLO is critical for the generation phagosomal pores, although eukaryotic cells may potentially experience a very limited amount of LLO at the PM; it is unclear whether the amounts used here are biologically relevant. Therefore, the mechanism of NM2A activation by PM pore formation may be a general mechanism to perturbation of the membrane or a change in membrane fluidity which is interesting but not necessarily a response to bacterial pathogenesis. These experiments would be more plausible as a response to bacterial infection, if a pore-forming toxin from a generally extracellular bacterium were used, for example PLY from Streptococcus pneumoniae or SLO from Streptococcus pyogenes. As presented, LLO and the Bacillus thuringiensis homolog are used as tools to induce PM pore formation but whether the identified NM2A activation pathway happens in the context of a true bacterial infection and not the addition of exogenous protein is unclear. Can the authors show evidence that NM2A activation happens in the context of Listeria monocytogenes infection of host cells or infections of C. elegans with Bacillus thuringiensis? Or does PLY or SLO trigger the same response observed from LLO?

Reviewer #3: 1. If the authors opt to maintain their argument that this Src-NMHC2A pathway is a response to LLO-induced damage; thus, informing us of host responses to Listeria pathogenesis, the authors should strongly consider repeating Figures 1AB or 1CD using host cells that were either not infected or infected with WT or Δhly Listeria. Furthermore, the use of a defective variant of either LLO (ex. LLOS176W) or other pore-forming toxins should be strongly considered to determine whether or not this signaling pathway is truly a response to the pore-forming activities of this toxin instead of alternative mechanisms (ex. contaminating endotoxins that trigger an immune response). Given the results on Figures 5C and 5E, whether this pathway is a general stress response to the activation of innate immune pathways instead of pore-formation itself should be considered.

2. The results of the propidium iodide assay, which tests the essentiality of NMHC2A phosphorylation status to PM repair using flow cytometry, demonstrate a modest effect size, which are potentially not statistically significant using a statistical test that takes into account unequal sample variances. Furthermore, there is quite a significant number of cells permeable to propidium iodide at baseline (~12% despite having no exposure to LLO), which may suggest improper gating or poor growth conditions of the cells. Lastly, as mentioned above, the use of defective variants for LLO or other pore-forming toxins should be considered to evaluate whether or not this signaling response is truly a response to “LLO-induced damage.” Thus, additional experiments and a more transparent account of the flow cytometry gating ought to be provided to allow readers to better assess the accuracy of the claim. Suggested supplementary material can include:

a. Provide the gating strategies for Figure 3E and include a positive control (ex. digitonin-permeabilized cells) to allow readers to identify which cells are truly positive for propidium iodide.

b. The use of transfected catalytically active or dead Src variants (as thoroughly provided in Fig. 2E) to assess positivity for propidium iodide (please see comment above to include control and gating strategies). Alternatively, use the NMHC2A variants in Brito, Pereira, Mesquita et al. 2023 (Ref. 24) to potentially complement information from Figure 3E.

c. Including a time-course of Figure 3E to document both positivity for propidium iodide over time and at increasing concentrations of LLO ( both WT and LLOS176W) and pair this with a measure of cell death (XTT, LDH, etc) assess whether or not components in this pathway truly contribute to PM repair and protection from pore forming toxins.

3. Finally, the biochemistry with the phophomutants is robust in demonstrating the lack of impact on ATPase activity and actin binding, however it is not clear that these point mutations lead to defects in the cellular response to LLO as assumed. These concstructs should be used to complement NMHC2A knockdown/knockout cells intoxicated with LLO to ensure that the model that these functions are not important for the response to LLO is valid.

**Part III – Minor Issues: Editorial and Data Presentation Modifications**

Reviewer #1: 1. Standardize nomenclature: the nematode specific toxin should be referred to as Cry5Ba (not Cry5B or Cry5b).

2. Clarify whether the authors assessed possible effects of NMHC2A phosphorylation on interactions with regulatory light chains or actin-binding partners.

3. The specificity of the Dasa inhibitor must be more rigorously addressed; otherwise, conclusions about Src dependency remain open to question.

4. The shRNA strategy raises concerns about off-target effects. Appropriate rescue experiments or additional validation would be needed to rule these out.

Reviewer #2: • Fig. 1, please include loading control quantification or normalization to the loading control

• Fig. 1C, please include a negative control such as a non-activated cell kinase

• Fig. 1B and 1D, please use consistent statistical analyses, specifically ANOVA tests

• Fig. 1E is slightly blurry and difficult to read

• Fig. 2, please include loading control quantification or normalization to the loading control

• Fig. 2B, 2D, 2F, and 2J please use consistent statistical analyses, specifically ANOVA tests

• Fig. 2G, the red and green words on the microscopy images is difficult to read

• Fig. 2G, the arrowed regions in NI are not clear

• Fig. 3A, please include the untreated control, shCtr (no LLO) control, and shSrc (no LLO) control, in addition to the quantification of the controls

• Fig. 5C and 5E, please use ANOVA for statistical analyses

• Fig. 5 results section, please include the rational for the heat stress assay

• Introduction, 2nd paragraph, “internalization vacuoles” should be “the phagosome”

• Pg. 7 typo “non-phophorylatable”

• Pg. 12 typos “activationand” and “cells boundaries”

• Discussion, please include additional information if available on other pore forming toxins and host cell pathway activation and whether this may be a general mechanism

Reviewer #3: 1. The integrated density measurements in Figure J alone does not match the degree of enrichment of pY158F in Figures 2GH. Perhaps a ratio of the integrated density in the cortical vs an immediately sub-cortical region (noting the consistent presence of pY158F signal in the nuclear/perinuclear region) can better reflect the enrichment of the signal at the cortical region.

2. The authors’ efforts in plotting individual replicates via scatter plots for nearly all quantifications are noted and applauded. Plotting standard deviation instead of SEM in the line graph in Figure 5B would similarly better allow the reader to assess the data variance for this key experiment.

PLOS authors have the option to publish the peer review history of their article (what does this mean? ). If published, this will include your full peer review and any attached files.

**Do you want your identity to be public for this peer review?** For information about this choice, including consent withdrawal, please see our Privacy Policy .

Reviewer #1: No

Reviewer #2: No

Reviewer #3: No

**Figure resubmission:**

**Reproducibility:**



---

## [Decision Letter · Decision Letter 1]

22 Jan 2026

PPATHOGENS-D-25-01957R1

Src-dependent tyrosine-phosphorylation of NM2A has a protective role against bacterial pore-forming toxins

PLOS Pathogens

Dear Dr. Sousa,

Thank you for submitting your manuscript to PLOS Pathogens. After careful consideration, we feel that it has merit but does not fully meet PLOS Pathogens's publication criteria as it currently stands. Therefore, we invite you to submit a revised version of the manuscript that addresses the points raised during the review process.

We look forward to receiving your revised manuscript.

Kind regards,

Francis Alonzo

Academic Editor

PLOS Pathogens

Michael Otto

Section Editor

PLOS Pathogens

Sumita Bhaduri-McIntosh

Editor-in-Chief

PLOS Pathogens

orcid.org/0000-0003-2946-9497

Michael Malim

Editor-in-Chief

PLOS Pathogens

orcid.org/0000-0002-7699-2064

**Additional Editor Comments:**

Thank you for submitting your revised article to PLoS Pathogens. Two of the original reviewers assessed the revisions to your manuscript. Both reviewers agreed that the work is much improved and have suggested only a few additional revisions to the text. The authors are encouraged to address the reviewer's suggestion for a softening of the language related to their conclusions, along with the minor addition of a relevant reference. Please see the reviewer comments for details.

**Journal Requirements:**

At this stage, the following Authors/Authors require contributions: Cláudia Brito, Francisco Sarmento Mesquita, Joana Maria Pereira, Daniel S. Osório, Neil Billington, Ricardo Rosário Lima, Sílvia Vale-Costa, James R. Sellers, Didier Cabanes, Ana X. Carvalho, and Sandra Sousa. Please ensure that the full contributions of each author are acknowledged in the "Add/Edit/Remove Authors" section of our submission form.

2) Please provide a complete Data Availability Statement in the submission form, ensuring you include all necessary access information or a reason for why you are unable to make your data freely accessible. If your research concerns only data provided within your submission, please write "All data are in the manuscript and/or supporting information files" as your Data Availability Statement.

3) Please amend your detailed Financial Disclosure statement. This is published with the article. It must therefore be completed in full sentences and contain the exact wording you wish to be published.

4) Please ensure that the funders and grant numbers match between the Financial Disclosure field and the Funding Information tab in your submission form. Note that the funders must be provided in the same order in both places as well.

5) Thank you for stating ' Full Western blot images and corresponding quantifications will be made publicly available via Zenodo. Additionally, all raw data used to generate the graphs in the manuscript will also be deposited and accessible through Zenodo.' Please note that, though access restrictions are acceptable now, your entire minimal dataset will need to be made freely accessible if your manuscript is accepted for publication. This policy applies to all data except where public deposition would breach compliance with the protocol approved by your research ethics board. If you are unable to adhere to our open data policy, please kindly revise your statement to explain your reasoning and we will seek the editor's input on an exemption. Please provide an active link to facilitate access to the data.

**Reviewers' Comments:**

Reviewer's Responses to Questions

**Part I - Summary**

Reviewer #2: In “Src-dependent tyrosine-phosphorylation of NM2A has a protective role against bacterial pore-forming toxins”, Brito et al. the authors identify an unrecognized pore forming toxin (PFT) induced Src-dependent pathway critical for cytoskeletal reorganization and response to plasma membrane damage. The authors use Caenorhabditis elegans to show a possible conserved mechanism in response to pore forming induced stress and heat shock. This is an interesting story and the authors have addressed several of the major issues described during the first review. The use of other PFTs (PLY and PFO) in addition to LLO showing they trigger NMHC2A Ty158 phosphorylation in a pore forming and calcium influx dependent manner make this paper more significant suggesting this may be a general response to PFTs. The authors have also used a Listeria monocytogenes deltaLLO control to demonstrate that LLO is critical for Src and tyrosine phosphorylation of NMHC2A. Over-all the manuscript is much improved compared to the first submission.

Reviewer #3: This revised manuscript has added significant improvements to address previous reviewer concerns. Notably the inclusion of data infecting with WT and ∆hly mutant Listeria and the addition of both other toxins and pore forming dead toxins significantly improve the argument that this signalling cascade is triggered by pore forming toxins and in the presence of bacterial infection. I still remain unconvinced that the authors have demonstrated membrane repair but rather have simply demonstrated differences in susceptibility to pore forming toxins which could be a result of increased pore formation and/or decreased pore repair. Language has been softened in the manuscript with regard to this point, however I think the authors could more clearly and explicitly state that their data is consistent with alterations in membrane repair but could also be indicative of increases in sensitivity of the cells. Finally, while I appreciate the authors attempting complementation with the biochemical mutants, the fact that these experiments were unsuccessful means that the authors need to be even more measured in their interpretation of the biochemical assays as they are simply unable to test whether these biochemical phenotypes are relevant in cells. In sum, while I believe the manuscript is significantly improved, these two aspects of the interpretation of the results are still somewhat overstated and could be softened to allow for the ambiguity that some of the data still contains.

**Part II – Major Issues: Key Experiments Required for Acceptance**

Reviewer #2: Over-all the authors have addressed my major concerns during the first submission. More specifically, the authors use sub-lethal amounts of LLO to show biological relevance, they use a Listeria monocytogenes deltaLLO control during infection, and they use other PFTs to show a general mechanism.

Reviewer #3: (No Response)

**Part III – Minor Issues: Editorial and Data Presentation Modifications**

Reviewer #2: Please describe and provide a citation for Draq7, in the section where Draq7 is used.

Otherwise, the authors have addressed my previous minor concerns.

Reviewer #3: (No Response)

PLOS authors have the option to publish the peer review history of their article (what does this mean? ). If published, this will include your full peer review and any attached files.

**Do you want your identity to be public for this peer review?** For information about this choice, including consent withdrawal, please see our Privacy Policy .

Reviewer #2: No

Reviewer #3: No

**Figure resubmission:**
---

## [Editor Report · Decision Letter 2]

28 Jan 2026

Dear Dr. Sousa,

We are pleased to inform you that your manuscript 'Src-dependent tyrosine-phosphorylation of NM2A has a protective role against bacterial pore-forming toxins' has been provisionally accepted for publication in PLOS Pathogens.

Best regards,

Francis Alonzo

Academic Editor

PLOS Pathogens

Michael Otto

Section Editor

PLOS Pathogens

Sumita Bhaduri-McIntosh

Editor-in-Chief

PLOS Pathogens

orcid.org/0000-0003-2946-9497

Michael Malim

Editor-in-Chief

PLOS Pathogens

orcid.org/0000-0002-7699-2064
---

## [Editor Report · Acceptance letter]

Dear Dr. Sousa,

We are delighted to inform you that your manuscript, "Src-dependent tyrosine-phosphorylation of NM2A has a protective role against bacterial pore-forming toxins," has been formally accepted for publication in PLOS Pathogens.

Best regards,

Sumita Bhaduri-McIntosh

Editor-in-Chief

PLOS Pathogens

orcid.org/0000-0003-2946-9497

Michael Malim

Editor-in-Chief

PLOS Pathogens

orcid.org/0000-0002-7699-2064